# A multiple super-enhancer region establishes inter-TAD interactions and controls *Hoxa* function in cranial neural crest

Sandra Kessler [1,2,4], Maryline Minoux[1,3,4], Onkar Joshi[1,4], Yousra Ben Zouari [1,4], Sebastien Ducret[1], Fiona Ross[1,2], Nathalie Vilain[1], Adwait Salvi[1,2], Joachim Wolff[1], Hubertus Kohler[1], Michael B. Stadler [1] & Filippo M. Rijli [1,2] ✉

Enhancer-promoter interactions preferentially occur within boundary-insulated topologically associating domains (TADs), limiting inter-TAD interactions. Enhancer clusters in linear proximity, termed super-enhancers (SEs), ensure high target gene expression levels. Little is known about SE topological regulatory impact during craniofacial development. Here, we identify 2232 genome-wide putative SEs in mouse cranial neural crest cells (CNCCs), 147 of which target genes establishing CNCC positional identity during face formation. In second pharyngeal arch (PA2) CNCCs, a multiple SE-containing region, partitioned into *Hoxa* Inter-TAD Regulatory Element 1 and 2 (HIRE1 and HIRE2), establishes long-range inter-TAD interactions selectively with *Hoxa2*, that is required for external and middle ear structures. HIRE2 deletion in a *Hoxa2* haploinsufficient background results in microtia. HIRE1 deletion phenocopies the full homeotic *Hoxa2* knockout phenotype and induces PA3 and PA4 CNCC abnormalities correlating with *Hoxa2* and *Hoxa3* transcriptional downregulation. Thus, SEs can overcome TAD insulation and regulate anterior *Hoxa* gene collinear expression in a CNCC subpopulation-specific manner during craniofacial development.

The establishment of complex spatiotemporal gene expression programs, controlling appropriate patterning events during development, involves regulatory DNA enhancer elements engaged in physical contact with their target gene promoters, sometimes over long distances[1]. The importance of these *cis*-regulatory elements during development is highlighted by the fact that their disruption can lead to disease and congenital disorders in humans[2–4]. Furthermore, a number of studies underline the relevance of enhancers in morphogenesis[5–11].

Among the most complex processes during embryonic development is the morphogenesis of the craniofacial skeleton, which derives from the cranial neural crest cells (CNCCs), a multipotent cell population that arises dorsally in the forming neural tube and migrates to colonize the facial and pharyngeal prominences[12–15]. To generate craniofacial skeletal and cartilage structures with proper shape, size, and orientation, CNCCs need to acquire specific positional identities and patterning information. This is achieved through the coordinated action of transcription factors, whose proper expression in the distinct facial and pharyngeal CNCC subpopulations is established in response to position-specific environmental signals[13,15,16].

*Hox* genes encode conserved homeodomain transcription factors that in mammals are organized into four clusters (*Hoxa-d*)[17–19]. While the CNCCs colonizing the frontonasal process and first pharyngeal arch (PA1) do not express *Hox* genes, *Hox* genes are required to provide rostrocaudal positional identity to the hindbrain rhombomeres (R)[20,21] and CNCCs of PA2-PA4, providing each CNCC subpopulation with unique patterning information[13,15,20]. Among the four clusters,

[1]Friedrich Miescher Institute for Biomedical Research, Maulbeerstrasse 66, 4058 Basel, Switzerland. [2]University of Basel, Basel, Switzerland. [3]INSERM UMR 1121, Université de Strasbourg, Faculté de Chirurgie Dentaire, 8, rue Sainte Elisabeth, 67 000 Strasbourg, France. [4]These authors contributed equally: Sandra Kessler, Maryline Minoux, Onkar Joshi, Yousra Ben Zouari. ✉e-mail: filippo.rijli@fmi.ch

*Hoxa* genes play a predominant role in patterning skeletogenic CNCCs[22]. *Hoxa2* is the only *Hoxa* gene expressed in CNCCs of PA2[23,24]. In the mouse, inactivation of *Hoxa2* induces homeotic transformation of the PA2-derived skeletal elements into a subset of PA1-like, *Hox*-negative, structures[25–28]. *Hoxa2* is also necessary and sufficient for external ear morphogenesis and *Hoxa2/HOXA2* hypomorph/haploinsufficient mutations in both mouse and humans cause microtia[26–31]. Moreover, *Hoxa2* and *Hoxa3* synergistically pattern PA3 and PA4 CNCC derivatives[22]. However, very little is known about how *Hoxa2* expression is regulated in PA2 CNCCs during craniofacial development, with only a few proximal regulatory elements identified in mouse[20] for which no functional data are available to date.

At the genome-wide level, in vivo epigenomic mapping and transgenic assays have identified putative enhancers that are active in the craniofacial prominences of embryonic day (E) 10.5 and 11.5 mouse embryos[5,16]. However, the target genes transcriptionally regulated by these putative enhancers remain largely unknown. This is partly due to the lack of comprehensive maps of 3D chromatin organization coupling these enhancers to their target promoters. Chromosome Conformation Capture (3C)-based techniques[32] have allowed the characterization of 3D chromatin interaction networks leading to the identification of specific long-range regulatory elements for developmentally important genes[33–39]. Moreover, these techniques have revealed that enhancer-promoter interactions preferentially occur within topologically associating domains (TADs), evolutionary and developmentally largely invariant 3D chromatin structures[40–42].

Very few putative long-range craniofacial enhancers have been functionally tested by in vivo targeted deletion so far, which resulted in quite mild variations in the shape and size of the affected structures[5,7,9]. This indicated that regulatory redundancy may contribute to buffer potentially deleterious phenotypic effects of single enhancer mutations[10]. Enhancer clusters, also termed super-enhancers (SEs), have been identified genome-wide as large, highly active regulatory domains containing clusters of active enhancers in close linear proximity, ensuring high expression levels of target genes[43,44]. However, little is known about SEs and their target genes active during craniofacial development, as well as about their potential functional impact on in vivo gene regulation in the context of 3D chromatin topology.

Here, we aimed to systematically identify SEs that might play a role in craniofacial morphogenesis by controlling the transcriptional regulation of key genes involved in establishing CNCC subpopulation-specific transcriptional programs[16]. Using Hi-C[45] and Promoter-Capture Hi-C (PCHi-C)[37] assays, in association with chromatin immunoprecipitation followed by sequencing (ChIP-seq) and Assay for Transposase-Accessible Chromatin using sequencing (ATAC-seq) we identified 2232 putative SEs. 147 of these targeting transcription factor-coding genes involved in establishing post-migratory CNCC positional identities[16]. We then focused on a large genomic region containing two subregions of 175 kb (*Hoxa* Inter-TAD Regulatory Element 1, HIRE1) and 39 kb (*Hoxa* Inter-TAD Regulatory Element 2, HIRE2) composed of multiple SEs, localized about 1.07 and 1.33 Mb away from the *Hoxa2* locus, respectively. We show that HIRE1 and HIRE2 are highly conserved in mammals, including humans, and establish inter-TAD long-range interactions with *Hoxa2* selectively in PA2 but not PA1 CNCCs, skipping the 3′ (centromeric) TAD neighboring the anterior *Hoxa* cluster. CRISPR-mediated targeted deletion of HIRE1 in the mouse phenocopied the full homeotic *Hoxa2* knockout phenotype in PA2 CNCCs and additionally induced malformations in PA3 and PA4 CNCC-derived skeletal structures, correlating with transcriptional downregulation of both *Hoxa2* and *Hoxa3*. In contrast, targeted deletion of HIRE2 did not yield major alterations of CNCC-derived skeletal structures, suggesting functional redundancy, but nonetheless resulted in microtic (i.e., small and malformed) pinnae in adult mice when put on a *Hoxa2* haploinsufficient sensitized background.

Thus, a multiple SE-containing region can overcome TAD insulation and provide very long-range transcriptional regulation in a CNCC subpopulation-specific manner and ensure robust target gene expression levels during craniofacial development.

## Results

### Genome-wide identification of super-enhancers in post-migratory CNCC subpopulations

To identify SEs that might play regulatory roles in CNCC subpopulations of distinct developing facial prominences, we analyzed H3K27ac ChIP-seq datasets from E10.5 mouse CNCCs of *Hox*-negative frontonasal process (FNP), maxillary (Mx) and mandibular (Md) components of PA1, and of *Hoxa2*-expressing PA2[16] (Fig. 1a). We first merged the H3K27ac ChIP-seq reads from these four CNCC subpopulations for peak calling and then largely followed the workflow of the ROSE algorithm to identify SEs[44,46] (Methods). Briefly, individual H3K27ac peaks within 12.5 kb or less were merged into larger regions, and, in each CNCC subpopulation, the H3K27ac signal was quantified to distinguish SEs from typical enhancers[44] (Supplementary Fig. 1). We excluded merged regions that overlapped with promoters, to identify only distal enhancer elements. In total, we found 2232 putative SEs that were active in at least one of the four CNCC subpopulations (Supplementary Data 1).

We next focused on the 237 transcription factor-coding genes differentially expressed between FNP, Mx, Md, and PA2 CNCCs (i.e., "positional" transcription factors, previously identified in ref. 16) and assessed if they were targeted by one or more SE(s). Among the putative SEs described above, for each CNCC subpopulation, we selected the SEs that were connected by at least one significant interaction to the promoters of positional transcription factor coding genes that displayed expression levels of RPKM >2 (Methods). To this aim, we performed PCHi-C[37] in duplicate for each of the four CNCC subpopulations (Supplementary Fig. 2a). Briefly, we used biotinylated RNA bait probes targeting promoter regions (Methods) to selectively enrich for all distal genome-wide sequences interacting with promoters from a pool of 'all-to-all' genomic interactions generated by Hi-C, followed by high throughput paired-end sequencing and statistical analysis. For CNCC collection, we micro-dissected E10.5 FNP, Mx, Md, and PA2 prominences and isolated red fluorescent protein (RFP)-expressing CNCCs by fluorescence-activated cell sorting (FACS) (Figs. 1a, 2a and Supplementary Fig. 3) (Methods).

In total, 62 out of 237 positional gene promoters showed, in the CNCC(s) where they were expressed, at least one significant interaction with at least one restriction fragment overlapping with a putative SE region (Fig. 1b and Supplementary Data 2). Among these targets, we found genes coding for transcription factors whose mutations result in craniofacial abnormalities in humans and/or are involved in prominence-specific CNCC patterning in mouse, including *Msx1/2*, *Tfap2b*, *Pax3*, *Alx4*, *Six1/2*, *Alx1*, *Hoxa2*, *Pitx1*, *Barx1*, *Meis1/2*, *Dlx3*, and *Hand2* (Fig. 1b and Supplementary Data 2). Some promoters were linked to multiple SEs, while one SE interacted with two gene promoters, namely those of *Hes6* and *Twist2*. Overall, we found 147 putative SEs associated with a positional transcription factor-coding gene promoter in at least one CNCC subpopulation (Fig. 1b and Supplementary Data 2).

### Identification of inter-TAD super-enhancers targeting *Hoxa2*

*Hoxa2* is the only *Hoxa* gene expressed in the CNCCs of PA2[23,24] (Fig. 2b) and is required to pattern all PA2-derived skeletal and cartilaginous structures, including the pinna[25–28]. Notably, PCHi-C revealed that 5 putative SEs, SE1–5, selectively targeted *Hoxa2* in E10.5 PA2 CNCCs, where this gene is highly expressed, unlike in the other CNCC subpopulations (Figs. 1b, 3a). SE1–5 were all located in a genomic region at a very large distance (>1 Mb) 3′ (centromeric) from the *Hoxa2* promoter (Figs. 1b, 3a). To assess temporal SE targeting dynamics and

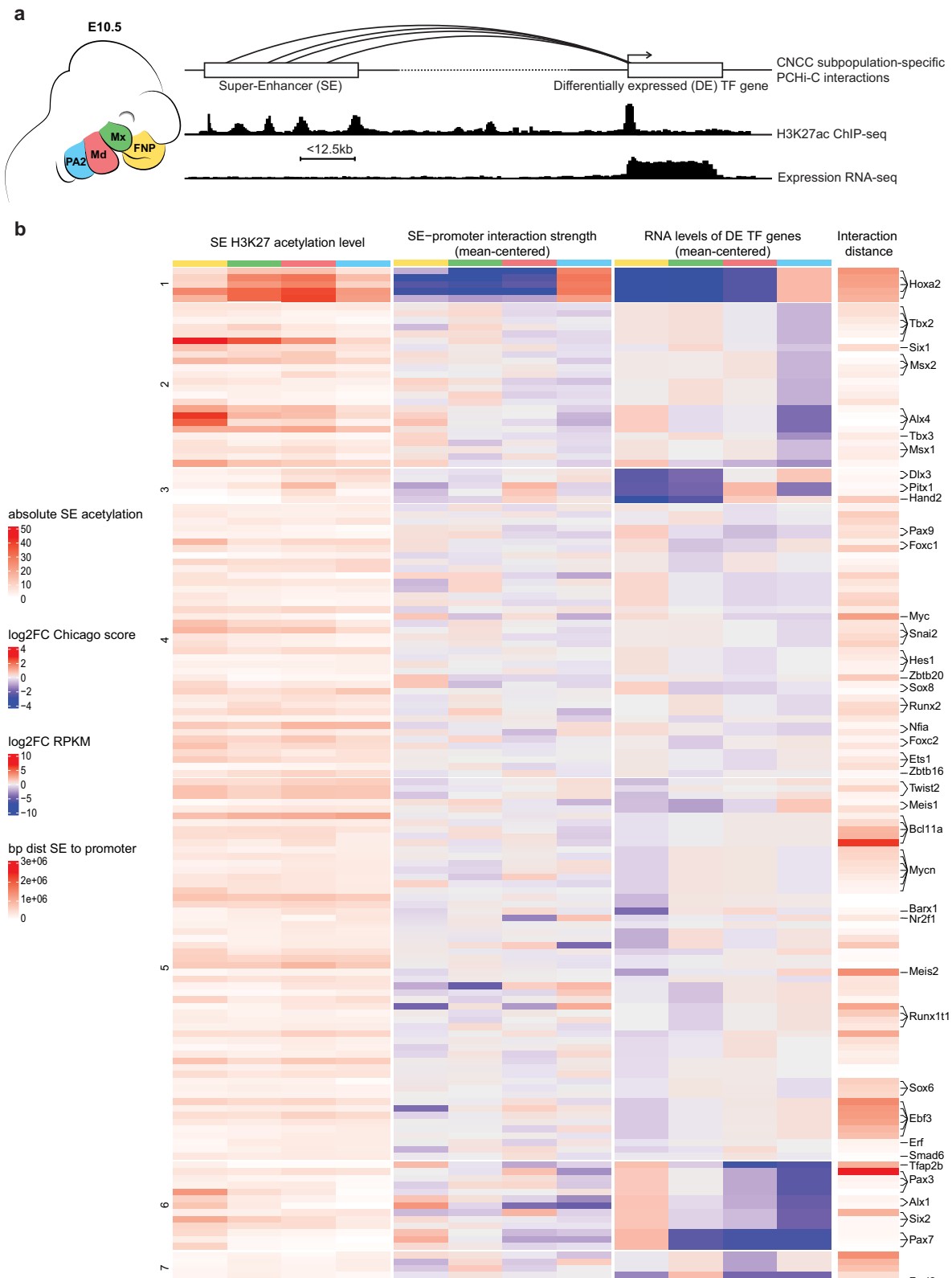

correlate it with the *Hoxa2* expression pattern in PA2 CNCCs and their derivatives, we analyzed PA2-derived pinna CNCCs at E12.5 and E14.5, in addition to E10.5 PA2 CNCCs. We micro-dissected E12.5 and E14.5 pinnae, isolated RFP-expressing CNCCs by FACS (Fig. 2a and Supplementary Fig. 3; Methods), and processed them for Hi-C, PCHi-C (Supplementary Fig. 2b), RNA-seq, ATAC-seq, and ChIP-seq assays. *Hoxa2* is highly expressed in E10.5 PA2 CNCCs[16] and its expression level further

increased in the E12.5 developing pinna (logFC = 0.753; FDR = 8.89E-07), while decreasing from E12.5 to E14.5 (logFC = −0.455; FDR = 5.71E-05) (Supplementary Data 3). Accordingly, the *Hoxa2* locus remained accessible and was enriched with active H3K27ac and H3K4me2 histone marks from E10.5 to E14.5, while the remainder of the transcriptionally silent *Hoxa* cluster was blanketed by the Polycomb (Pc)-dependent repressive H3K27me3 mark (Fig. 2b)[16]. In contrast, in *Hox-*

**Fig. 1 | Super-enhancer calling and assignment to positional transcription factor-coding genes. a** (Left) Schematic of mouse facial prominences at embryonic day 10.5 (E10.5). Cranial neural crest cells (CNCCs) of the frontonasal process (FNP), maxillary (Mx), mandibular (Md), and second pharyngeal arch (PA2) are depicted in yellow, green, red, and blue, respectively. (Right) Each CNCC subpopulation was subjected to RNA-seq, H3K27ac ChIP-seq, and PCHi-C. Promoter distal ChIP-seq peaks with a maximum distance of 12.5 kb from each other were used for super-enhancer (SE) calling. Patterns of ChIP-seq and RNA-seq signals at one SE-promoter pair, where both elements are active, are shown in the style of genome browser tracks. Links of SEs to their target genes were identified with PCHi-C focusing on positional transcription factor (TF) coding genes[16] and are represented as arcs. **b** Heatmap of the SEs assigned to positional transcription factor-coding genes. Each heatmap row represents a SE-promoter pair. In each CNCC subpopulation, the SEs

were linked to promoters of positional transcription factor coding genes[16] if there was at least one significant interaction between the two elements, and if the gene was expressed (>2 RPKM). 147 SEs were linked to 62 different genes, with 148 unique pairings. The row annotation highlights transcription factor coding genes involved in craniofacial development and/or malformations, if mutated. From left to right, the heatmap displays the H3K27 acetylation level of each SE for each CNCC subpopulation, the mean CHiCAGO score of all interactions from a promoter bait of a positional transcription factor coding gene to a restriction fragment that overlaps with a SE, the expression of the positional transcription factor coding genes and the distance between the two elements. The interaction strength and gene expression are given as log2 FC for each CNCC subpopulation in relation to the mean across all four CNCC subpopulations. The rows were grouped by k-means clustering on the gene expression levels (cluster numbers are indicated on the left).

free E10.5 Md CNCCs, the whole *Hoxa* cluster was embedded into a Polycomb H3K27me3 repressive domain (Fig. 2b)[16]. As in E10.5 PA2, in E12.5 and E14.5 pinna CNCCs, SE1–5 were accessible and active, as indicated by ATAC-seq peaks and H3K27ac enrichment (Fig. 3a and Supplementary Fig. 4a).

Hi-C profiles of mouse embryonic stem cells (mESCs) and E10.5 Md CNCCs, and of *Hoxa2*-expressing PA2 and pinna CNCCs at E10.5, E12.5, and E14.5 further showed that, in all these cell populations, the *Hoxa* cluster was embedded in a dense domain spanning the border between two adjacent 3′ and 5′ TADs[47] (blue box, Fig. 3b and Supplementary Fig. 2c). As assessed by CTCF binding profiles (Supplementary Fig. 2d) and TAD separation score (black arrow, Fig. 3b and Supplementary Fig. 2c; Methods), no difference among cell populations was observed in the position of a strongly predicted boundary, segregating the *Hoxa2* locus from the SE1–5-containing genomic region in distinct TADs (Fig. 3b and Supplementary Fig. 2c, d). SE1 partially overlapped this TAD boundary at its 5′ end, whereas SE2–5 covered a genomic region encompassing almost entirely the neighboring TAD across that boundary (Fig. 3b and Supplementary Fig. 2c).

Given their profile of interactions and the relative proximity of SE2–4 as compared to SE5, we further subdivided SE2–5 into two main regulatory subdomains. The first subdomain (chr6:50,913,170-51,087,888), hereafter referred to as '*Hoxa* inter-TAD regulatory element 1' (HIRE1) is located about 1.07 Mb away from *Hoxa2* and covers a 175 kb region encompassing SE2–4 (excluding the SE2 most 5′ end which does not interact with *Hoxa2*) (Fig. 3a, b and Supplementary Fig. 4). The second subdomain (chr6:50,789,172-50,828,639), referred to as HIRE2, is a 39 kb region encompassing SE5 and localized about 1.33 Mb away from *Hoxa2* (Fig. 3a, b and Supplementary Fig. 4). The SE1–5 interaction patterns with *Hoxa2* in PA2-derived CNCCs were similar from E10.5 through E14.5 (Fig. 3a, b). However, HIRE1 interacted more strongly with *Hoxa2* in PA2 and pinna CNCCs at E10.5 and E12.5, whereas HIRE2 had the strongest interaction in pinna CNCCs at E12.5 (Fig. 3c, d and Supplementary Fig. 5a). Notably, in *Hox*-negative Md, Mx, and FNP CNCC subpopulations, even though not interacting, HIRE1 and HIRE2 were also accessible and enriched with the active H3K27ac mark (Fig. 3 and Supplementary Fig. 4a), whereas the *Hoxa2* locus was maintained repressed by Polycomb-dependent H3K27me3 (Fig. 2b)[16].

The TAD containing HIRE1 and HIRE2 is a gene-poor chromosomal region (Fig. 3c). Virtual 4C plots from the HIRE1 or HIRE2 viewpoints confirmed the *Hoxa2*-specific interactions. Furthermore, they identified weak intra- and inter-TAD interactions with *Npvf* and *Hoxa1/Hoxa3*, respectively (Supplementary Fig. 5). None of these genes are expressed in the selected CNCC subpopulations, suggesting that these weak contacts are not functional nor specific, likely due to physical proximity of *Hoxa1/Hoxa3* to *Hoxa2* and *Npvf* to the HIRE1/2 SEs, respectively. Moreover, most Hi-C-based methods cannot resolve very proximal interactions (typically <30 kb) from the generally high background crosslinking frequency between genomic sequences over these distances. Thus, very long-range inter-TAD interactions were

established between *Hoxa2* and HIRE1/HIRE2 in E10.5 PA2 and E12.5-E14.5 pinna CNCCs. Remarkably, *Hoxa2* inter-TAD interactions were visible as an asymmetrical "architectural stripe"[48] on low-resolution Hi-C plots (arrowheads, Fig. 3b and Supplementary Fig. 5a) and confirmed at higher resolution by PCHi-C (Fig. 3a). Inter-TAD interactions were selective for PA2 and pinna CNCCs, which express *Hoxa2* (Figs. 2b, 3a, b and Supplementary Fig. 2c), as they were absent in *Hox*-free mESCs and E10.5 Md CNCCs (Fig. 3a, b and Supplementary Fig. 2c).

## HIRE1 and HIRE2 are highly conserved in mammals

HIRE1 and HIRE2 sequences were highly conserved in eutherian mammals, whereas in more phylogenetically distant species, such as marsupials, birds, or fish, the degree of conservation was considerably lower (Fig. 4). Nevertheless, both HIRE2 and HIRE1 contained multiple 200–1000 base pair (bp) long elements showing high conservation in basewise analysis across 60 vertebrates. In HIRE1, 22 of such elements were conserved in bird species, with two elements notably conserved in fish as well (Fig. 4a). In HIRE2, six elements were conserved to some degree down to birds and two out of six showed conservation in the coelacanth fish (Fig. 4b). Furthermore, these highly conserved elements tended to overlap with ATAC-seq peaks in developing CNCCs and derivatives (Fig. 4, blue bars), suggesting a conserved functional role of HIRE1 and HIRE2 in vertebrates, in particular, Eutheria.

## HIRE1 deletion phenocopies the full *Hoxa2* knockout phenotype, whereas HIRE2 deletion in a *Hoxa2* haploinsufficient background results in microtia

To assess their involvement in *Hoxa2* transcriptional regulation, we deleted HIRE1 and HIRE2 using the CRISPR-Cas9 system (Fig. 5a and Supplementary Fig. 4b; Methods). Mice lacking one copy of HIRE2 (*HIRE2^{del/wt}*) appeared phenotypically normal. Similarly, as compared to wild-type (WT), E18.5 *HIRE2^{del/del}* homozygous mutant fetuses (*n* = 6) did not display visible abnormalities of CNCC-derived pinna (Fig. 5b, c), middle ear, or hyoid structures (Fig. 6a–f and Supplementary Fig. 6a–d). To address the potential impact of the HIRE2 deletion on a haploinsufficient *Hoxa2* background, we generated trans-heterozygous mutants (Fig. 5a) by mating *HIRE2^{del/wt}* mice with *Hoxa2^{EGFP/wt}* mice, carrying a *Hoxa2* knockout allele[49]. While no clear defect of the pinna was visible at E18.5 in *HIRE2^{del/wt};Hoxa2^{EGFP/wt}* fetuses (*n* = 4; 8 pinnae) (Fig. 5d), adult trans-heterozygous animals displayed microtia, i.e., smaller and misshapen ears (Fig. 5e–g, *n* = 5/5; 10 pinnae). Mild skeletal abnormalities could be observed at E18.5 (Fig. 6g–i and Supplementary Fig. 6e, f), namely, the PA2-derived processus brevis of the malleus was reduced (asterisk, Fig. 6g, h, *n* = 8/8 sides) and an ectopic cartilage nodule was inconsistently present on the PA2-derived styloid process (white arrow, Fig. 6g, *n* = 5/8 sides). This suggests that HIRE2 is mostly functionally redundant, although still required to contribute to full *Hoxa2* expression levels (see below) on a sensitized *Hoxa2* haploinsufficient genetic background.

HIRE2 deletion might be mainly functionally compensated by HIRE1. Heterozygous mutant mice for HIRE1 deletion (*HIRE1^{del/wt}*)

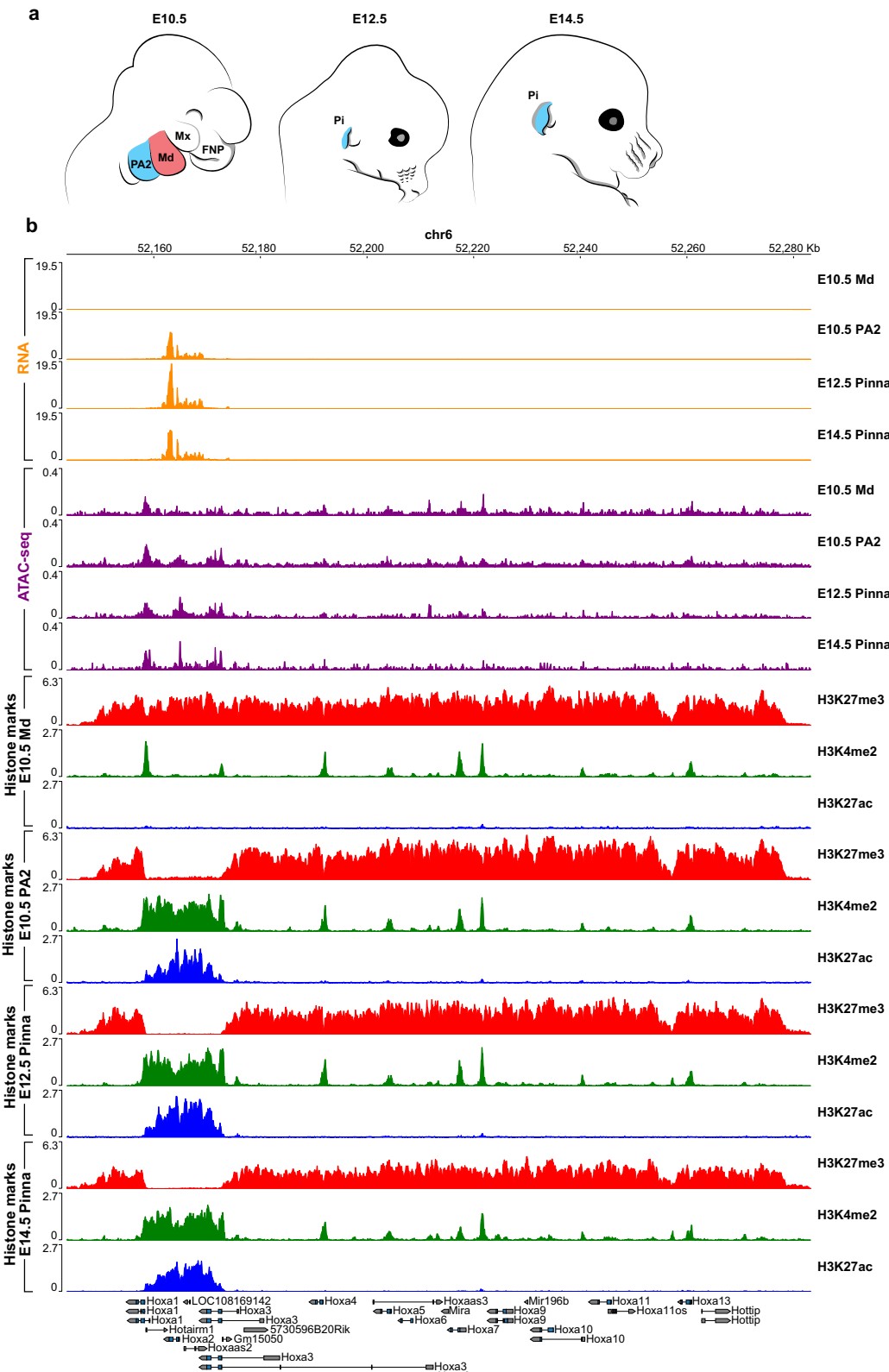

**Fig. 2 | Chromatin state at *Hoxa2* locus. a** Schematic of mouse developmental progression from E10.5 to E14.5. The mandibular process (Md) at E10.5 is highlighted in red. The second pharyngeal arch (PA2) at E10.5 and the PA2-derived pinna (Pi) at E12.5 and E14.5 are highlighted in blue. Mx, maxillary process of first pharyngeal arch; FNP, frontonasal process. **b** *Hoxa* cluster genome browser view with RNA (orange), chromatin accessibility (ATAC-seq, purple), and ChIP-seq profiles for H3K27me3 (red), H3K4me2 (green), and H3K27ac (blue) from E10.5 Md and PA2 cranial neural crest cells (CNCCs), and E12.5 and E14.5 pinna CNCCs.

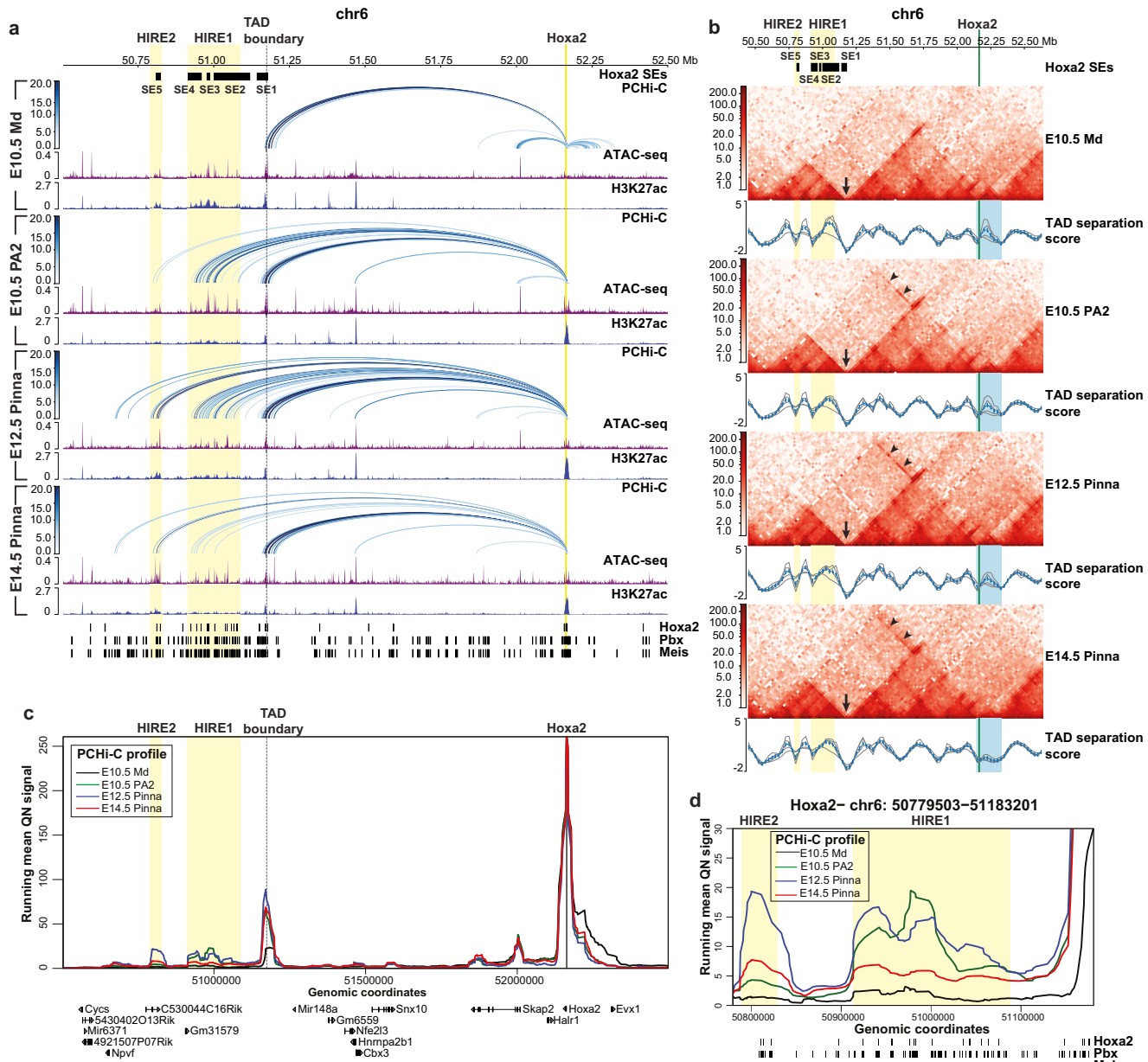

**Fig. 3 | *Hoxa* long-range inter-TAD regulatory elements (HIREs). a** Genome browser view of significant promoter capture Hi-C (PCHi-C) interactions (blue arcs) for *Hoxa2*, chromatin accessibility (ATAC-seq, purple), and ChIP-seq profile for H3K27ac (blue) in the mandibular process (Md) and second pharyngeal arch (PA2) cranial neural crest cells (CNCCs) at E10.5, and in E12.5 and E14.5 pinna CNCCs in a 2 Mb region of chromosome 6 (50,502,806–52,500,000 bp). PCHi-C interactions with a CHiCAGO score ≥5 are visualized. The color intensity of an arc indicates the CHiCAGO score, with a maximum value of 20 (i.e., interactions having a score ≥20 are shown in dark blue). Only interactions with *Hoxa2* are shown. Bottom, ChIP-seq binding sites for Hoxa2[57], Pbx, and Meis[58] in PA2 at E11.5. **b** Hi-C interaction heat-maps at 25 kb resolution in a 2.2 Mb region of chromosome 6, including the *Hoxa* cluster (50,442,417–52,636,150 bp) in Md and PA2 CNCCs at E10.5, and in pinna CNCCs at E12.5 and E14.5. TAD separation scores are called with HiCExplorer's hicFindTADs (shown as blue lines). Additional gray lines show the TAD scores for different window sizes. The blue highlight marks the domain encompassing the *Hoxa* cluster and *Evx1* (chr6:52,145,433–52,327,518). Arrows indicate the location of a TAD boundary between *Hoxa2* and HIRE1/HIRE2. Arrowheads highlight inter-TAD interactions between *Hoxa2* and HIRE1/HIRE2. **c** Virtual 4C profiles derived from PCHi-C of E10.5 Md (black) and PA2 (green) CNCCs and pinna CNCCs at E12.5 (blue) and E14.5 (red) on the *Hoxa2* promoter bait. The same chromosomal region is displayed for panels **a**, **c**. For better readability, genomic coordinates are only displayed below panel **c** and within the *Hoxa* cluster only show *Hoxa2* genomic position. **d** Zoom in on virtual 4C profiles of panel **c** at HIRE1 and HIRE2 (chr6:50779503 – 51183201). Bottom, ChIP-seq binding sites for Hoxa2[57], Pbx, and Meis[58] in PA2 at E11.5. In **a**–**d**, HIRE1 and HIRE2 are highlighted by yellow boxes. The *Hoxa2* locus is highlighted in yellow (**a**) or green (**b**). Black boxes at the top of panels **a** and **b** show the *Hoxa2* super-enhancer 1–5 (SE1–5) in PA2 CNCCs at E10.5.

appeared normal. In contrast, E18.5 *HIRE1^del/del^* homozygous mutant (n = 4) and *HIRE1^del/wt^;Hoxa2^EGFP/wt^* trans-heterozygous mutant (n = 4) fetuses survived up to birth but died perinatally. All *HIRE1^del/del^* and *HIRE1^del/wt^;Hoxa2^EGFP/wt^* mutants lacked the pinna, similarly to the full *Hoxa2^−/−^* mutant phenotype[26] (Fig. 5h, i). Moreover, all *HIRE1^del/del^* (n = 4/4, 8 sides) and *HIRE1^del/wt^;Hoxa2^EGFP/wt^* (n = 4/4, 8 sides) mutants

displayed skeletal malformations phenocopying the full *Hoxa2^−/−^* phenotype[26]. Namely, the PA2 CNCC-derived stapes, styloid process and lesser horns of the hyoid bone were absent and replaced by a mirror image homeotic duplication of PA1-like structures, including a duplicated incus, malleus, tympanic bone, and a partially duplicated Meckel's cartilage (Fig. 6j–o). Like *Hoxa2^EGFP/EGFP^* mutants[49] (n = 6/8

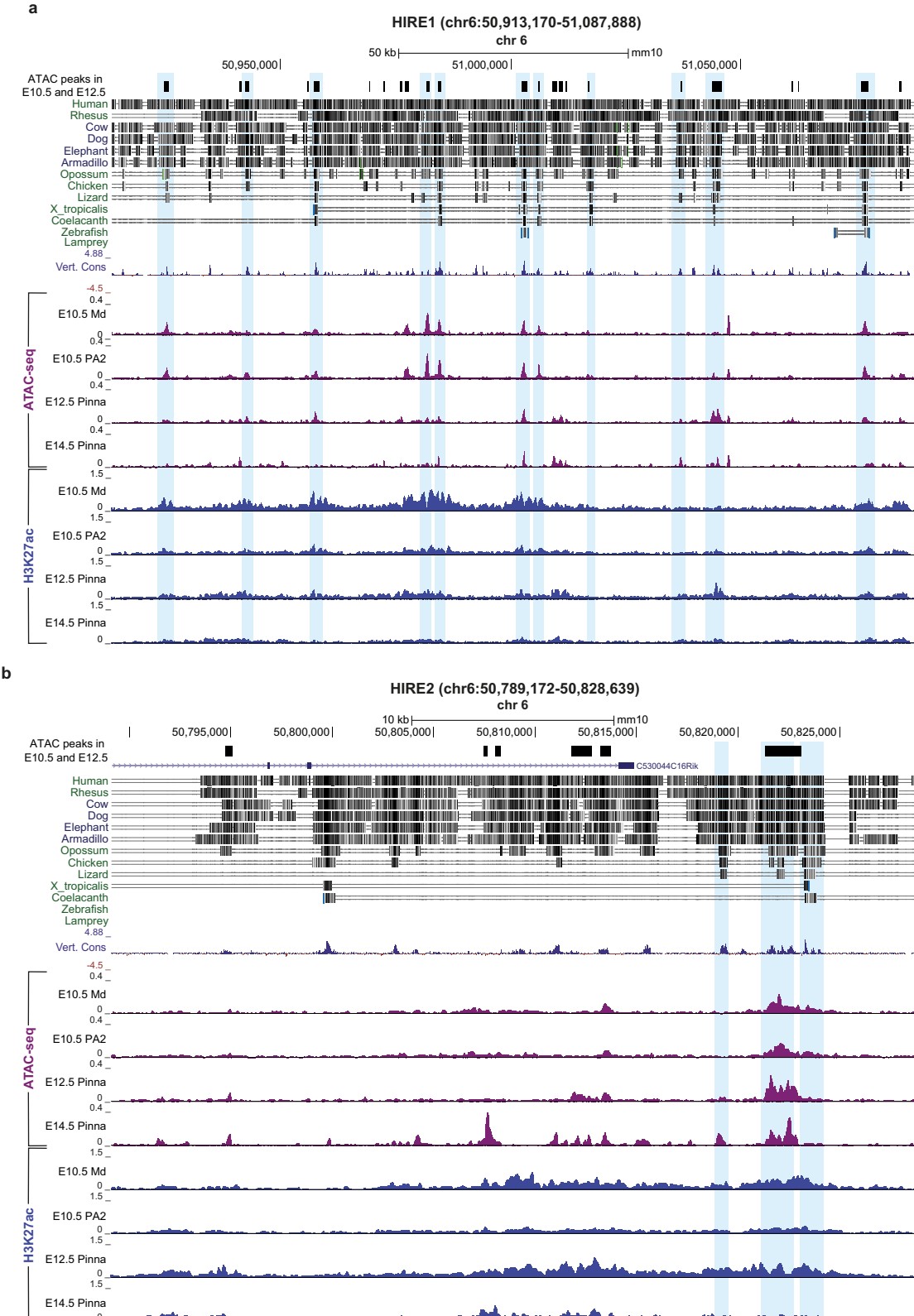

**Fig. 4 | Conservation of HIRE1 and HIRE2 in Eutheria/Placentalia.** Illustration of sequence conservation at HIRE1 (**a**) and HIRE2 (**b**). Top, combined ATAC-seq peaks called for E10.5 second pharyngeal arch (PA2) cranial neural crest cells (CNCCs) and E12.5 pinna CNCCs. The related sequence alignments of the indicated vertebrate species to the mouse genomic DNA sequence by Multiz program[134] are shown, as well as the 60 vertebrates Basewise Conservation (Vert. Cons) by PhyloP (blue/red), both extracted from UCSC genome browser. Corresponding chromatin accessibility (purple) and H3K27ac pattern (blue) in mandibular (Md) and PA2 CNCCs at E10.5 and pinna CNCCs at E12.5 and E14.5 are depicted. The blue boxes highlight regions with high levels of conservation between mouse and other vertebrate species at HIRE1 and HIRE2, which also show ATAC-seq peak overlap in CNCCs at different developmental stages.

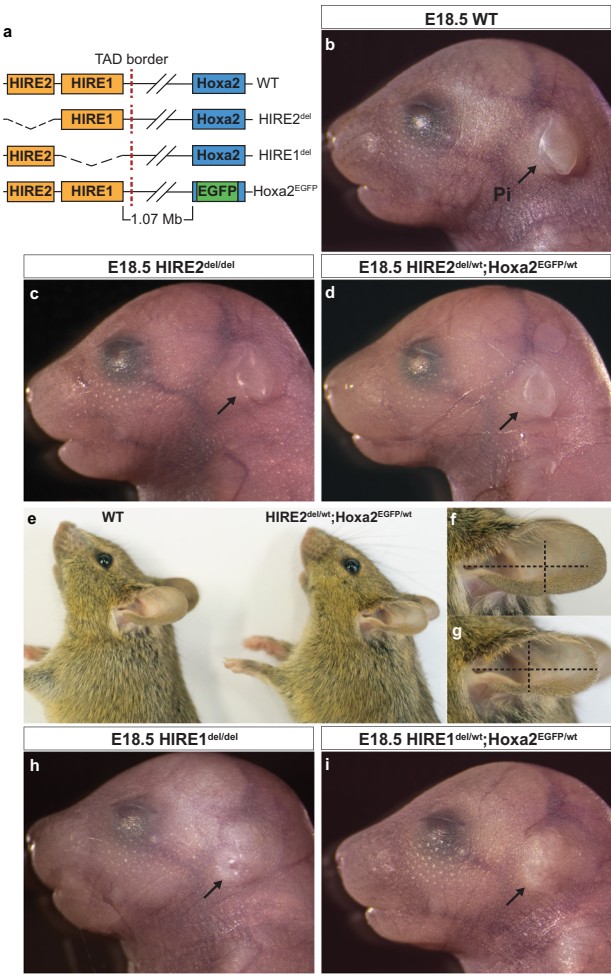

**Fig. 5 | Effect of HIRE1 and HIRE2 deletions on pinna morphogenesis. a** CRISPR/Cas9 mediated deletion of HIRE1 and HIRE2 in vivo. In the *Hoxa2^EGFP* knockout allele, *Hoxa2* is replaced by *EGFP* knock-in[49]. **b–d** E18.5 external ear phenotype in wild-type (WT) (**b**, representative of *n* = 4/4 fetuses), *HIRE2^del/del* homozygous mutant (**c**, representative of *n* = 6/6 fetuses) and *HIRE2^del/wt;Hoxa2^EGFP/wt* trans-heterozygote mutant (**d**, representative of *n* = 4/4 fetuses). **e** External ear phenotype in WT and *HIRE2^del/wt;Hoxa2^EGFP/wt* adult mice. **f, g** Enlarged views of the pinna of WT (**f**, representative of *n* = 4/4 animals) and *HIRE2^del/wt;Hoxa2^EGFP/wt* (**g**, representative of *n* = 5/5 animals) adult mice in **e. h, i** E18.5 external ear phenotype in *HIRE1^del/del* homozygous mutant (**h**, representative of *n* = 4/4 fetuses) and *HIRE1^del/wt;Hoxa2^EGFP/wt* trans-heterozygous mutant (**i**, representative of *n* = 4/4 fetuses). Arrows show the pinna (Pi), which has no visible abnormalities in **c** and **d** as compared to **b** but is absent in **h** and **i** (phenocopying the *Hoxa2^-/-* phenotype[26]). The vertical and horizontal dashed lines of equal length in panels **f** and **g** highlight the shape and size differences of the external ear between WT and *HIRE2^del/wt;Hoxa2^EGFP/wt* adult mice.

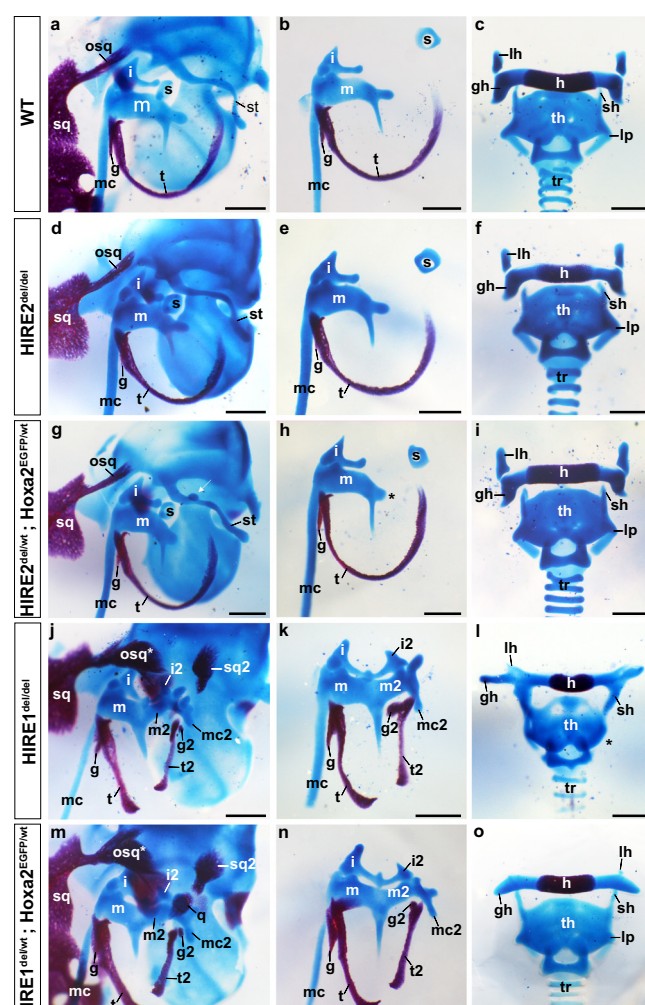

**Fig. 6 | Middle ear and hyoid skeletal changes in HIRE1 and HIRE2 mutant fetuses.** Middle ear (**a, b, d, e, g, h, j, k, m, n**) and hyoid (**c, f, i, l, o**) skeletal preparations from E18.5 wild-type (WT) (**a–c**), *HIRE2^del/del* homozygous (**d–f**), *HIRE2^del/wt;Hoxa2^EGFP/wt* trans-heterozygous (**g-**), *HIRE1^del/del* homozygous (**j–l**), and *HIRE1^del/wt;Hoxa2^EGFP/wt* trans-heterozygous (**m–o**) mutant fetuses. *HIRE2^del/del* skeletal structures appear normal (**d–f**). **a–n** are representative images of WT (*n* = 8/8) (**a, b**), *HIRE2^del/del* (*n* = 12/12) (**d, e**), *HIRE2^del/wt;Hoxa2^EGFP/wt* (*n* = 8/8) (**g, h**), *HIRE1^del/del* (*n* = 8/8) (**j, k**) and *HIRE1^del/wt;Hoxa2^EGFP/wt* (*n* = 8/8) (**m, n**) fetus sides. **c, f, i, l, o** are representative images of WT (*n* = 4/4) (**c**), *HIRE2^del/del* (*n* = 6/6) (**f**), *HIRE2^del/wt;Hoxa2^EGFP/wt* (*n* = 4/4) (**i**), *HIRE1^del/del* (*n* = 4/4) (**l**), and *HIRE1^del/wt;Hoxa2^EGFP/wt* (*n* = 4/4) (**o**) fetuses. *HIRE2^del/wt;Hoxa2^EGFP/wt* mutant fetuses have a smaller processus brevis (\*, **h**) and can display a cartilage nodule on the styloid process (st) (white arrow, **g**, *n* = 5/8 fetus sides) compared to WT (**a, b**). In *HIRE1^del/del* (**j, k**) and *HIRE1^del/wt;Hoxa2^EGFP/wt* (**m, n**) middle ear homeotic duplications phenocopying those of conventional *Hoxa2^-/-* mutant[26] are observed. In *HIRE1^del/wt;Hoxa2^EGFP/wt*, the lesser horns (lh) of the hyoid bone (h) are absent and greater horns (gh) display an abnormal location, similar to *Hoxa2^-/-* fetuses[26] (**o**). Superior horns (sh) of thyroid cartilage (th) are elongated, and lateral processes (lp) of laryngeal cartilage are reduced (**o**). In *HIRE1^del/del* mutants, lh are absent, gh extends dorsally, sh fuse with gh (**l**) and lp are absent (\*, **l**). g and g2 represent WT and duplicated gonial bones, respectively; i and i2, WT and duplicated incus, respectively; m and m2, WT and duplicated mallei, respectively; mc and mc2, WT and partially duplicated Meckel's cartilages, respectively; osq and osq\*, normal and modified retrotympanic (otic) process of squamosal bone; s, stapes; sq, squamosal bone; sq2, ectopic squamosal bone; t and t2, WT and duplicated tympanic bones, respectively; tr trachea. Scale bars represent 500 μm.

fetuses), both *HIRE1^del/del* and *HIRE1^del/wt;Hoxa2^EGFP/wt* mutants also displayed a cleft secondary palate (*n* = 8/10 and 14/24 fetuses, respectively) (Supplementary Fig. 7). Furthermore, additional typical features of *Hoxa2^-/-* mutant fetuses could be observed in both *HIRE1^del/del* and *HIRE1^del/wt;Hoxa2^EGFP/wt* mutants, such as partial duplications of the pterygoid and squamosal bones, and bifurcation of the retrotympanic process of the orthotopic squamosal bone (Supplementary Fig. 6g–j), strongly suggesting that HIRE1 contributes to most of *Hoxa2* expression in PA2 CNCCs (see below).

We nonetheless observed variability in the extent of morphological transformation of ectopic elements, as compared to *Hoxa2^-/-* phenotype, namely in the shape of duplicated malleus and Meckel's cartilage (Fig. 6k, n and Supplementary Fig. 8a–c; *n* = 8/8 sides for *HIRE1^del/del* and *n* = 2/8 sides for *HIRE1^del/wt;Hoxa2^EGFP/wt* mutants). In some cases, there was no fusion between the incus and its duplicated

counterpart (Supplementary Fig. 8a, b; *n* = 1/8 sides for *HIRE1^del/del* and *n* = 1/8 sides for *HIRE1^del/wt;Hoxa2^EGFP/wt*) and/or in most cases the gonial bone was mirror-image duplicated (Fig. 6j, k, m, n and Supplementary Fig. 8b, c; *n* = 8/8 sides for *HIRE1^del/del* and *n* = 6/8 sides for

*HIRE1 del/wt ;Hoxa2 EGFP/wt*) whereas in full *Hoxa2−/−* mutants the duplicated gonial bone was fused to its normal counterpart into a single element[26] (Supplementary Fig. 8a). Moreover, the lesser horns of the hyoid bone were sometimes highly reduced, instead of absent (Fig. 6o and Supplementary Fig. 8d). Altogether, these mild phenotype variations could be explained by differences from the original genetic background and/ or the presence of residual *Hoxa2* transcripts (see below).

A notable difference between *HIRE1 del/del* and *HIRE1 del/wt ;Hoxa2 EGFP/wt* mutants concerned the ectopic atavistic palatoquadrate skeletal structure, described in *Hoxa2−/−* mutants[26]. This structure was present with full penetrance in *HIRE1 del/wt ;Hoxa2 EGFP/wt* trans-heterozygous, though never in *HIRE1 del/del* mutants (Supplementary Fig. 6g–j). Moreover, in E18.5 *HIRE1 del/wt ;Hoxa2 EGFP/wt* trans-heterozygous mutants, the PA3-derived greater horns of the hyoid bone made an abnormal angle with the body of the hyoid bone, similar to the *Hoxa2−/−* phenotype[26] (Fig. 6o and Supplementary Fig. 8d, g), the superior horns of the thyroid cartilage were elongated, and the lateral processes of the laryngeal cartilage were reduced (Fig. 6o and Supplementary Fig. 8d). In contrast, in all *HIRE1 del/del* homozygous mutants a fusion between the superior horns of the thyroid cartilage and the greater horns of the hyoid bone could be observed (Fig. 6l and Supplementary Fig. 8e), a phenotype reminiscent of *Hoxa3−/−* knockout mutants[50,51]. In addition, the lateral process of the thyroid cartilage was absent (asterisk, Fig. 6l and Supplementary Fig. 8e) and the greater horns of the hyoid bone extended dorsally and fused with the dorsal part of the thyroid cartilage (*n* = 5/8) (Supplementary Fig. 8h, i). These malformations are reminiscent, while not identical, of those observed upon *Hoxa* cluster deletion in CNCCs[22]. As *Hoxa4* is not expressed at significant levels in PA3 and PA4[22,52], these malformations could therefore result from simultaneous downregulations of both *Hoxa2* and *Hoxa3* expression (see below). Hence, in addition to regulating *Hoxa2* expression in PA2 CNCCs, HIRE1 might be required to regulate *Hoxa2* and *Hoxa3* expression in PA3 and PA4 CNCCs. This is further emphasized by the finding that the observed malformations of PA3 and PA4-derived structures are stronger in *HIRE1 del/del* than in *HIRE1 del/wt ;Hoxa2 EGFP/wt* mutants, where the removal of only one allele of HIRE1 would result in a less severe reduction of *Hoxa3* expression.

The skeletal abnormalities in *HIRE1 del/del* and *HIRE1 del/wt ;Hoxa2 EGFP/wt* mutants were restricted to CNCC-derived structures. The axial and limb skeletons were normal. Moreover, while the otic capsule lacked the oval window as in full *Hoxa2−/−* mutants[26], there was no otic skeletal phenotype reminiscent of *Hoxa1−/−* mutants[53,54], suggesting that *Hoxa1* expression is not regulated by HIRE1.

## HIRE1 is required for *Hoxa2* and *Hoxa3* expression in CNCCs

To assess the decrease in *Hoxa2* expression following the inactivation of HIRE1 or HIRE2, we performed a qRT-PCR analysis (Methods) in PA2 of E10.5 *HIRE2 del/del*, *HIRE2 del/wt ;Hoxa2 EGFP/wt*, *HIRE1 del/del* and *HIRE1 del/wt ;Hoxa2 EGFP/wt* mutant embryos (Supplementary Fig. 9). We collected PA2 of E10.5 WT and *Hoxa2 EGFP/wt* embryos as controls (Supplementary Fig. 9). In *Hoxa2 EGFP/wt* embryos, which do not display skeletal malformations, we detected about 50% of the normal *Hoxa2* transcript levels, as compared to WT (Supplementary Fig. 9). *HIRE2 del/del* and *HIRE2 del/wt ;Hoxa2 EGFP/wt* mutant embryos displayed 75% and 46% of the normal *Hoxa2* transcript levels, as compared to WT, respectively. This correlates with the absence of visible malformations of PA2-derived structures in E18.5 *HIRE2 del/del* mutant fetuses, and the presence of mild malformations in E18.5 *HIRE2 del/wt ;Hoxa2 EGFP/wt* mutant fetuses (Figs. 5, 6 and Supplementary Figs. 9, 10). By contrast, in *HIRE1 del/del* and *HIRE1 del/wt ;Hoxa2 EGFP/wt* mutant embryos, we found a drastic reduction of *Hoxa2* expression, with about 0–3% and 0–4% of *Hoxa2* transcripts, respectively, as compared to WT, thus explaining the finding that HIRE1 deletion phenocopied the full homeotic *Hoxa2* knockout phenotype in PA2 CNCCs (Figs. 5, 6 and Supplementary Figs. 9, 10).

Next, we carried out whole-mount and tissue section in situ hybridization for *Hoxa2* and *Hoxa3* in *HIRE1 del/del* embryos. At E10.5, *Hoxa2* expression was undetectable in PA2, in keeping with the qRT-PCR data (Supplementary Fig. 9), and highly reduced in more posterior pharyngeal arches (Fig. 7a, c, d, f and Supplementary Fig. 11a, b), as well as in the anterior hindbrain R2 to R5, as compared to WT (Fig. 7b, e). *Hoxa2* expression was, however, maintained normally in somites (Fig. 7a, d). *Hoxa3* expression was highly reduced in PA3 and more posterior pharyngeal arches (Fig. 7g, i, j, l and Supplementary Fig. 11c, d) but only slightly affected in the hindbrain of E10.5 *HIRE1 del/del* embryos (Fig. 7h, k) and maintained normally in somites (Fig. 7g, j). We did not detect changes in the expression patterns of *Hoxa1*, which at E8.5 is expressed in the hindbrain (Supplementary Figs. 11e–k, 8f–l).

Next, to quantify the transcriptional changes induced by HIRE1 deletion on both *Hoxa2* and *Hoxa3* and on their potential downstream targets in the pharyngeal region, we dissected PA2 and PA3 in E10.5 WT and *HIRE1 del/del* embryos and performed RNA-seq. In keeping with the in situ hybridization results, *Hoxa2* and *Hoxa2/Hoxa3* transcript levels were significantly reduced in PA2 and PA3, respectively, of *HIRE1 del/del* embryos as compared to WT (Fig. 8 and Supplementary Data 3). *Hoxa2* downregulation was, however, more severe in PA2 than in PA3 (Fig. 8a, c and Supplementary Data 3), while in PA3, *Hoxa2* relative transcript levels decreased less than *Hoxa3*, as compared to WT (Fig. 8a, c, e and Supplementary Data 3). Thus, the effect of HIRE1 on *Hoxa* gene regulation correlates with the gene linear position in the cluster. Secondly, confirming in situ hybridization experiments, our data indicate a stronger effect of HIRE1 on *Hoxa2* transcriptional regulation in its anterior-most domain of expression, both in the pharyngeal region and hindbrain.

Furthermore, in PA2 of E10.5 *HIRE1 del/del* mutants, 16 genes were upregulated and 41 downregulated, excluding *Hoxa2* itself (FDR <0.05 and log2 CPM ≥1), as compared to WT (Fig. 8a, b and Supplementary Data 3). In PA3, 8 genes were upregulated and 21 downregulated, excluding *Hoxa2* and *Hoxa3* (FDR <0.05 and log2 CPM ≥1) (Fig. 8c, d and Supplementary Data 3). Among the upregulated genes, we confirmed several genes known to be negatively regulated by Hoxa2, including *Gbx2, Pitx1, Alx4, Lhx6, Barx1*, and *Rspo2*[27,55–57] (Fig. 8a, b and Supplementary Data 3). Among the downregulated genes, we identified several known Hoxa2 targets, such as *Meis1, Meis2, Meox1, Fzd4, Zfp703, and Zfp503*[56–60] (Fig. 8a, b and Supplementary Data 3). These data confirm the specific role of HIRE1 for *Hoxa2* and *Hoxa3* transcriptional regulation in PA2 and PA3.

We next investigated if HIRE1 could also be required at the time of emergence and early stages of CNCC migration. In the mouse, *Hoxa2* expression in PA2 CNCCs is detected at an early migratory stage around E8.25-E8.5[23,24,61]. Notably, in *HIRE1 del/del* embryos, *Hoxa2* transcripts were already undetectable by in situ hybridization at E8.5 in the CNCCs arising from R4 and migrating into PA2 (Fig. 7m, q); no signal was detected at E9.0 and E9.5 as well (Fig. 7n–p, r–t), similarly to E10.5 *HIRE1 del/del* embryos (Fig. 7a–f). Moreover, we observed a strong *Hoxa2* downregulation in the anterior hindbrain of *HIRE1 del/del* embryos already at E8.5, from R2 to R5 (Fig. 7m, q), and even more so at E9.0 and E9.5 (Fig. 7n–p, r–t).

Altogether, our findings strongly suggest that HIRE1 is required for induction of high *Hoxa2* expression levels already at the emergence and earliest stages of CNCC migration and may also be involved in maintaining appropriate transcript levels through later developmental stages, in both hindbrain and CNCCs.

## Identification of transcription factor binding motifs in HIRE1/ HIRE2 and involvement of *Hoxa2* in its own long-range regulation

To investigate which transcription factors may be involved in binding HIRE1 and HIRE2 and potentially regulate long-range interactions with

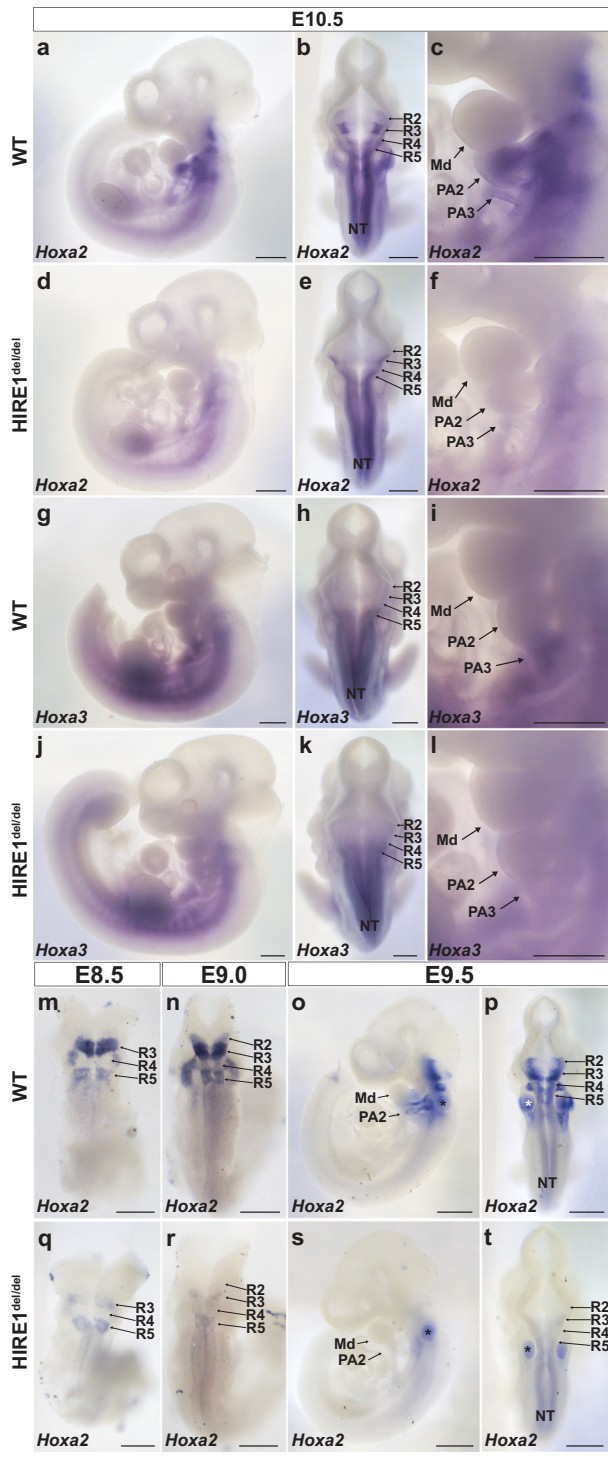

**Fig. 7 | HIRE1 is required for *Hoxa2* and *Hoxa3* expression.** Whole-mount in situ hybridization on E10.5 wild-type (WT) (**a**–**c**, **g**–**i**) and *HIRE1*[del/del] (**d**–**f**, **j**–**l**) embryos using *Hoxa2* (**a**–**f**) and *Hoxa3* (**g**–**l**) antisense probes (**a**, **c**, **d**, **f**, **g**, **i**, **j**, **l**, **o**, **s**) lateral and (**b**, **e**, **h**, **k**, **m**, **n**, **p**, **q**, **r**, **t**) dorsal views. **a**–**c** are representative images of E10.5 WT embryos (*n* = 6), **d**–**f** are representative images of E10.5 *HIRE1*[del/del] embryos (*n* = 6), **g**–**i** are representative images of E10.5 WT embryos (*n* = 3), and **j**–**l** are representative images of E10.5 *HIRE1*[del/del] embryos (*n* = 3). *Hoxa2* expression is not detectable in second (PA2) and third (PA3) pharyngeal arches of *HIRE1*[del/del] embryos (**d**, **f**) and is reduced in hindbrain rhombomeres 2–5 (R2–R5) (**e**) compared to WT (**a**–**c**). *Hoxa3* expression is strongly reduced in PA3 of *HIRE1*[del/del] embryos (**j**, **l**) and not affected in the hindbrain (**k**) compared to WT (**g**–**i**). Whole-mount in situ hybridization on WT (**m**–**p**) and *HIRE1*[del/del] (**q**–**t**) E8.5 (**m**, **q**), E9.0 (**n**, **r**), and E9.5 (**o**, **p**, **s**, **t**) embryos with *Hoxa2* antisense probe. **m**–**t** are representative images of *n* = 3 embryos for each stage and genotype. In E8.5 and E9.0 *HIRE1*[del/del] mutant embryos, *Hoxa2* expression is undetectable in R2 and R4 and severely reduced in R3 and R5 (**q**, **r**) compared to WT (**m**, **n**). At E9.5, *Hoxa2* expression is not detectable in PA2 of *HIRE1*[del/del] embryos (**s**) and is strongly reduced in hindbrain R2-5 (**t**) compared to WT (**o**, **p**). *, trapped dye in the otic capsule. Md, mandibular process. In **a**–**i**, **o**, **p**, **s**, **t** scale bars represent 500 μm. In **m**, **n**, **q**, **r** scale bars represent 250 μm.

Next, assuming that the observed accessibility changes are driven by differential transcription factor binding and that other genomic regions bound by the same transcription factors would show similar accessibility profiles, we ranked all ATAC peaks in E10.5 PA2 and E12.5 pinna CNCCs according to their similarity (Pearson's correlation coefficient) to the average accessibility profile of each of the three clusters (Supplementary Fig. 12c). We selected the 1000 peaks with the highest correlation to each cluster profile, resulting in three non-overlapping peak sets (Supplementary Fig. 12d). We then used these three sets of peaks and, as a control, a fourth set containing all residual ATAC peaks, and ran a motif enrichment analysis[62] resulting in a total of 382 significantly enriched motifs (Supplementary Fig. 12e and Supplementary Data 4). From these, we hierarchically clustered similar motifs and selected nine representative motifs with predicted transcription factor binding sites in the 31 ATAC-seq peaks overlapping with the HIRE2/HIRE1 region (Fig. 9a, Supplementary Data 5, 6, and Supplementary Figs. 12g, 13) (Methods).

For example, among the representative motifs, there were binding sites for Tal- and NFAT-related factors, potentially bound by Twist1, Twist2, and ZBTB18 and Nfatc1, Nfat5, and Nfatc3, respectively (Fig. 9a and Supplementary Fig. 12g). Each of these transcription factors is upregulated from E10.5 to E12.5 in CNCCs (Supplementary Data 3). Furthermore, we identified motifs for Hox-related factors, such as Hoxa2, and TALE-type homeodomain factors, such as Pbx and Meis. Both Meis and Pbx transcription factors are known Hox cofactors and form heterodimers with Hox proteins to bind to DNA[63,64]. Analysis of published ChIP-seq datasets for Hoxa2, Pbx, and Meis in PA2 at E11.5[57,58], revealed that these factors indeed showed enriched binding at HIRE1 and HIRE2 (Fig. 3a and Supplementary Fig. 4).

These latter findings indicated that Hoxa2 may play a role in long-range inter-TAD interactions with its own promoter. Notably, PCHi-C on E10.5 PA2 CNCCs in *Hoxa2*[EGFP/EGFP] knockout embryos[49] revealed that the strength of the interactions of *Hoxa2* with HIRE2 in E10.5 *Hoxa2*[EGFP/EGFP] knockout embryos was reduced by 27% (indeed, note that with CHiCAGO score ≥5, no arc is visible between *Hoxa2* and HIRE2 in *Hoxa2*[EGFP/EGFP] vs. WT; Fig. 9d and Supplementary Fig. 3). These data strongly suggest that Hoxa2 is partially involved in its own long-range transcriptional regulation likely with cofactors such as Pbx and Meis.

## Discussion

Large clusters of enhancers in close genomic proximity and collectively bound by arrays of transcription factors, termed SEs, have been involved in the transcriptional control of cell identity during differentiation and disease[43,44,65–67]. Less is known about the potential

*Hoxa2*, we performed a motif enrichment analysis using the ATAC-seq data from PA2 and pinna CNCCs at E10.5 and E12.5, respectively. We first called the peaks from the datasets of both stages, merged them, and extracted putative enhancer peaks to obtain a total of 106,587 peaks. Thirty-one ATAC-seq peaks overlapped the region spanning from HIRE2 to HIRE1 (mouse GRCm38/mm10 chr6:50,789,172–51,087,888) (Supplementary Fig. 12a). We then clustered these 31 peaks according to their relative accessibility at E10.5 and E12.5 (cluster 1–3) (Supplementary Fig. 12b). Cluster 1 peaks were more accessible at E12.5, cluster 2 peaks were more accessible at E10.5, whereas cluster 3 peaks only showed minor accessibility differences between E10.5 and E12.5.

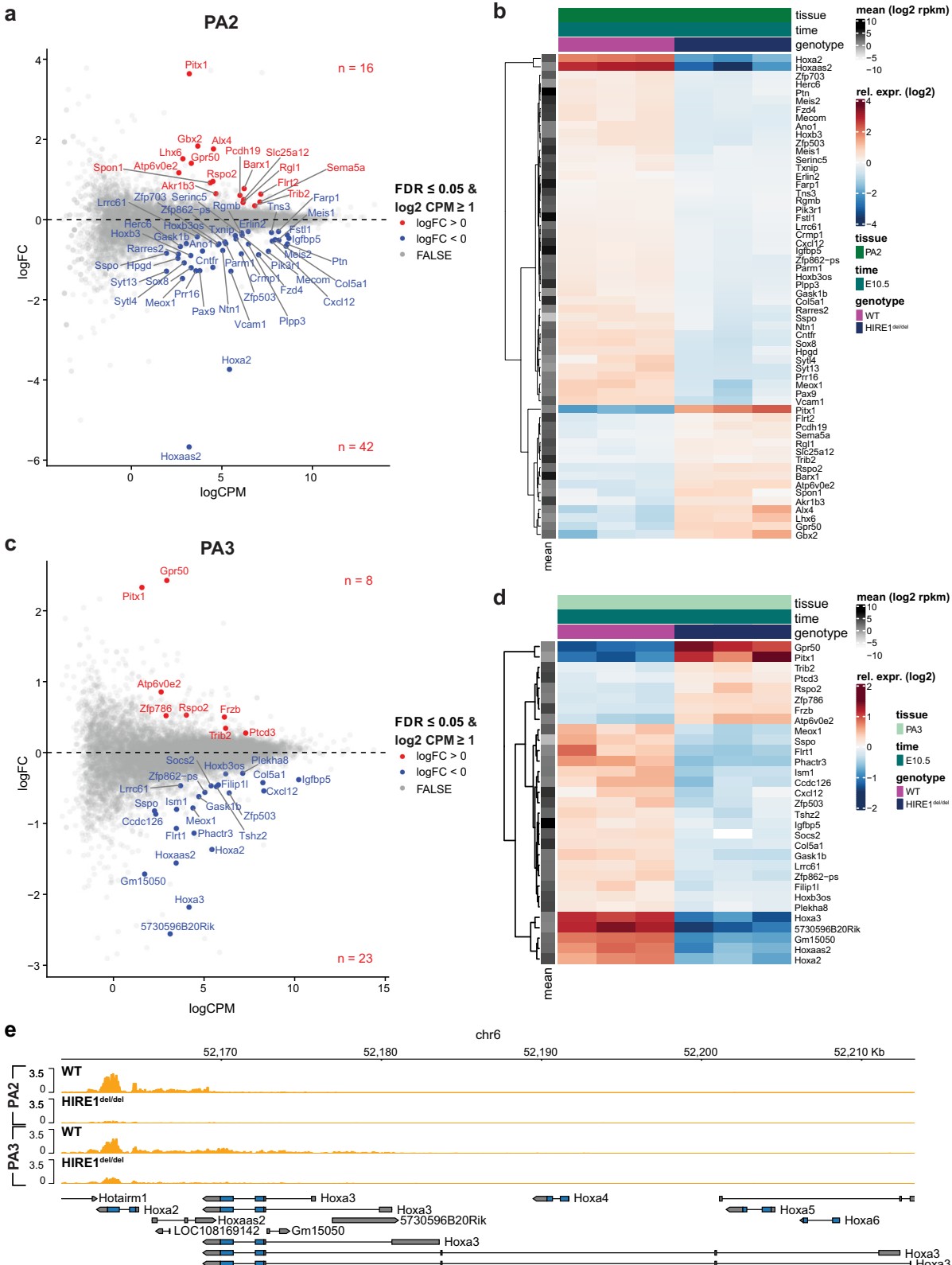

**Fig. 8 | Genome-wide transcriptional effects of HIRE1 deletion.** MA plots (**a**, **c**) showing the log2 fold-change (FC) versus the average log2 count per million (CPM) for the differential gene expression between wild-type and E10.5 *HIRE1^del/del^* embryos, in pharyngeal arch 2 (PA2) (**a**) and 3 (PA3) (**c**). Each dot represents a single gene. Differentially expressed genes with FDR ≤0.05 and log2 CPM ≥1 are shown as red and blue dots. Positive log2 FC values correspond to genes upregulated (red dots) and negative log2 FC corresponds to genes downregulated (blue dots) in *HIRE1^del/del^*. Heatmaps (**b**, **d**) of RNA levels relative to the mean over all wild-type and *HIRE1^del/del^* replicates in PA2 (**b**) and PA3 (**d**) at E10.5. Differentially expressed genes with FDR ≤0.05 and log2 CPM ≥1 are shown. (**e**) Genome browser view at *Hoxa2* and *Hoxa3* loci showing RNA profiles (orange) in PA2 and PA3 of E10.5 wild-type (WT) and *HIRE1^del/del^* embryos.

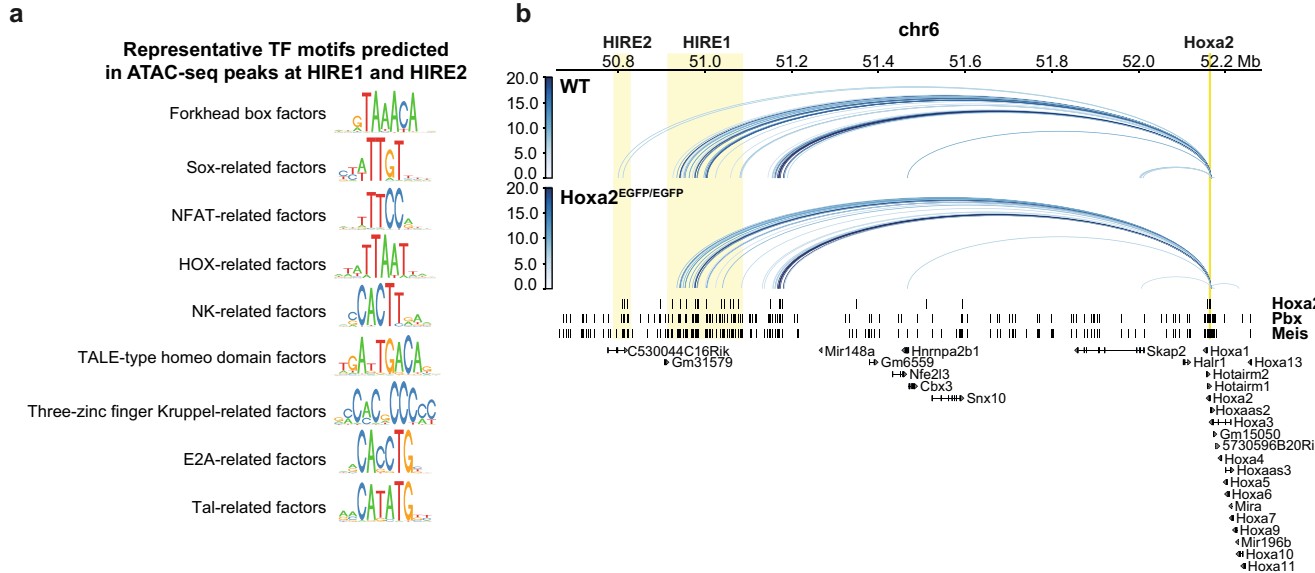

**Fig. 9 | Involvement of Hoxa2 in inter-TAD long-range interactions between its promoter and HIRE2. a** Representative transcription factor (TF) motifs predicted to be associated with ATAC-seq peaks of E10.5 PA2 and E12.5 pinna cranial neural crest cells (CNCCs) at HIRE1 and HIRE2. Each of the depicted motifs is representative of a cluster of similar motifs (Methods, Supplementary Figs. 12g, 13). **b** Significant promoter capture Hi-C interactions for *Hoxa2*, shown as blue arcs, in second pharyngeal arch (PA2) CNCCs of wild-type (WT) and *Hoxa2$^{EGFP/EGFP}$* embryos at E10.5, in a 1.7 Mb region of chromosome 6 (50,656,605–52,283,516). As in Fig. 3, only interactions with a CHiCAGO score ≥5 are visualized. The color intensity of an arc indicates the CHiCAGO score, with a maximum value of 20 (i.e., interactions having a score ≥20 are shown in dark blue). The visualized interactions are filtered to show only *Hoxa2* interactions. Significant interactions from *Hoxa2* to HIRE2 are lost in *Hoxa2$^{EGFP/EGFP}$*. Bottom, binding sites, identified by ChIP-seq, for Hoxa2[57], Pbx, and Meis[58] in PA2 at E11.5. HIRE1, HIRE2, and *Hoxa2* locus are highlighted by yellow boxes.

involvement of SEs in establishing the positional identities of specific cell populations during embryo morphogenesis. During craniofacial development, distinct migratory CNCC subpopulations contributing to the skeletal structures building the face acquire their positional identities and patterning information by the differential expression of key transcription factors induced by local signals and drive specific transcriptional programs in the different facial and pharyngeal prominences[15]. Here, we computationally identified 2232 putative CNCC subpopulation-specific SEs. Using PCHi-C, we found that 147 out of 2232 SEs selectively targeted 62 positional transcription factor-coding genes previously shown to be differentially expressed between FNP, Mx, Md, and PA2 CNCC subpopulations during craniofacial development[16]. Most interestingly, we identified very long-range (>1 Mb) inter-TAD SEs (SE1-SE5) that interact with *Hoxa2* in a CNCC subpopulation-specific manner. Based on their proximity, we grouped tandem SE2–4 into HIRE1, a 175 kb geneless genomic region, while SE5 encompassed a 39 kb region, termed HIRE2. HIRE1 and HIRE2 are highly conserved in mammals, including humans. Single (HIRE2, SE5) or multiple (HIRE1, SE2-SE4) functional deletions of inter-TAD SE elements in the mouse resulted in skeletal phenotypes with distinct severity and penetrance, partially or fully phenocopying *Hoxa2* knockout (Supplementary Fig. 10) as well as aspects of the *Hoxa3* mutant skeletal phenotypes. We further show that inter-TAD SEs selectively control the expression of *Hoxa2* and *Hoxa3* in neural derivatives, such as CNCCs and hindbrain rhombomeres, though not in mesoderm-derived tissues.

The putative craniofacial SEs, including *Hoxa2* SE1-SE5, were identified using the ROSE algorithm[44,46] (Methods). Similar to the original SE definition[44], we operationally used a maximum distance of 12.5 kb between individual active enhancers to consider them as part of a SE. This threshold is, however, arbitrary. Thus, even though SE2-SE4/HIRE1 and SE5/HIRE2 are separated by 84.4 kb without strong enrichment in H3K27ac, they might nevertheless be considered to belong to a single, extremely large (spanning 300–400 kb), distant SE regulatory

region targeting *Hoxa2* (and *Hoxa3*) in neural derivatives during development.

During facial morphogenesis, the transcriptional output of key genes needs to be tightly regulated, as certain structures may be sensitive even to small perturbations of gene dosage resulting in patterning abnormalities and disease (e.g., lower jaw[7]). *Hoxa2* provides positional identities and patterning information to all PA2 CNCCs derivatives and its function is highly conserved in vertebrate CNCCs[25,26,68–71]. In the mouse, *Hoxa2* is mainly required for morphogenesis of external (pinna) and middle ear structures[25,26,28,72]. Spatiotemporal control of *Hoxa2* expression levels is important to pattern distinct CNCC-derived skeletal elements[27,73,74], but how this is regulated at the transcriptional level is poorly understood. Enhancer clusters or SEs acting on a common target gene could provide a suitable regulatory landscape to control and fine-tune transcriptional output during the morphogenesis of facial elements. Previous work has shown that, in the presence of multiple enhancer clusters or SEs, individual enhancer constituents within clusters, or even individual enhancer clusters, may have overall weak activity on the target promoter and act in a partially redundant manner, when deleted; however, simultaneous deletion of multiple (clusters of) enhancers may result in synergistic combinatorial effects on target gene transcription levels[7,75–77]. The finding that *Hoxa2* expression is selectively regulated in the CNCCs of PA2 and derivatives by the activity of multiple clusters of inter-TAD long-range enhancers (SE1-SE5) prompted us to analyze their functional role by CRISPR-Cas9 mediated deletion.

Due to its partial overlapping with the TAD boundary, functional analysis of SE1 was not further pursued to avoid the potentially confounding effects of its deletion. HIRE2 homozygous inactivation did not result in a detectable *Hoxa2* mutant phenotype, indicating that its deletion could be fully compensated and redundant with other enhancers and/or identified SEs. In humans, HOXA2 haploinsufficiency causes microtic ears and hearing impairment[29–31], indicating that external ear morphogenesis is sensitive to HOXA2 dosage reduction

already at 50% of its normal levels. Remarkably, when put on a sensitized *Hoxa2* haploinsufficient genetic background, the deletion of HIRE2 was also associated with smaller, microtic, external ears in adult mice (Fig. 5e–g). Accordingly, HIRE2 is more enriched with H3K27ac and interacts more strongly with *Hoxa2* in pinna CNCCs at E12.5 (i.e., at the beginning of its formation) than in E10.5 PA2 (Fig. 3 and Supplementary Fig. 4). Thus, while HIRE2 homozygous deletion causes a modest reduction of *Hoxa2* expression (Supplementary Figs. 9, 10), which is nonetheless still compatible with full *Hoxa2* function, the HIRE2/SE5 contribution to *Hoxa2* transcription and function becomes critical below a 50% expression threshold, affecting the morphogenesis of dosage-sensitive structures such as the pinna (Supplementary Figs. 9, 10).

Strikingly, HIRE1 deletion phenocopied the full homeotic *Hoxa2* knockout phenotype, including the lack of pinna, and, in addition, resulted in *Hoxa3* knockout-like phenotypes in PA3-PA4-derived skeletal structures (Fig. 6j–o and Supplementary Figs. 6e–h, 7, 8). In *HIRE1^{del/del}* mutants, we confirmed strong downregulation of both *Hoxa2* and *Hoxa3* in CNCCs and anterior hindbrain (Figs. 7, 8), indicating that these distant SEs are involved in collinear regulation of *Hoxa2* and *Hoxa3* in neural derivatives and supporting the observation that SEs can coordinately regulate clustered genes[75,78]. The much stronger effect of HIRE1, as compared to HIRE2, most likely reflects the fact that HIRE1 comprises multiple SEs, namely SE2, SE3, and SE4, while HIRE2 is only composed of SE5. Even though HIRE1 deletion results in a complete lack of pinna, under dosage-sensitive conditions, HIRE2 does contribute to pinna morphogenesis as well (see above). This strongly indicates partial redundancy among SEs to achieve robustness against genetic or environmental perturbations, as well as cooperative or synergistic contributions to boost *Hoxa2* transcription and ensure reproducible expression patterns driving harmonious morphogenesis. These hypotheses will need to be tested by additional in vivo deletions and analysis.

Multiple cis-regulatory elements proximal to, or within, the *Hoxa2* locus have been identified, driving reporter genes expression in R2 and somites[79–81], R3/R5[61,82–84], R4[79,85,86], as well as in CNCCs[61]. However, the in vivo functional role of these proximal enhancers and their contribution to endogenous *Hoxa2* expression and patterning activity was not investigated. One of the major findings of this study is that most of the *Hoxa2* transcriptional output in neural crest and rhombomeres is dependent on multiple long-range, inter-TAD, interactions with clustered regulatory elements, HIRE1 and HIRE2, spanning a large genomic region of more than 1 Mb away from *Hoxa2*. To date, only a handful of very long-range regulatory sequences at more than 1 Mb genomic distance from their promoters have been identified, and they are all intra-TAD-located. Namely, these include the *Shh* ZRS, *Myc* BENC and MNE, and *SOX9* EC1.45 and EC1.25 enhancers[7,9,87–89]. This is the first time that an inter-TAD (super-)enhancer genomic region was identified and shown to be critical for the regulation of a key vertebrate developmental gene during face morphogenesis.

We generated E10.5, E12.5, and E14.5 CNCC subpopulation-specific Hi-C and PCHi-C data and CTCF ChIP-seq binding profiles and showed that the TAD organization flanking the *Hoxa* cluster is similar in Md and PA2 CNCCs at all stages analyzed and in ES cells (Fig. 3b and Supplementary Fig. 2c, d). Importantly, we identified a strong invariant TAD boundary with identical CTCF binding patterns in Md and PA2 CNCCs, that partitioned *Hoxa2* from HIRE1 and HIRE2 in distinct TADs in both cell populations (Fig. 3b and Supplementary Fig. 2d). This prompted the question of how these SEs can communicate in a subpopulation-specific manner with the *Hoxa2* promoter across this TAD boundary. Even though the functional relationship between TAD topology and gene regulation is debated[90], most of the enhancer-promoter (E-P) interaction pairs reside within the same TAD, suggesting that CTCF-bound TAD borders, while not absolutely required for intra-TAD E-P interactions[91], may provide intra-TAD transcriptional insulation thus

limiting inter-TAD E-P interactions[1,90]. On the other hand, TAD identification depends on Hi-C data resolution and the algorithm used for TAD calling[92]. Based on such computational methods, inter-TAD E-P interactions detected at a cell population level are indeed relatively rare[37,93,94]. However, single-cell approaches revealed greater than expected cell-to-cell heterogeneity and dynamic behavior of TADs, with only about 1.5–2.0-fold enrichment of intra-TAD vs. inter-TAD interactions in single cells[95]. Moreover, recent work[96] showed that enhancer strength, boundary strength, and distance all determine promoter sensitivity to CTCF-mediated transcriptional insulation at TAD boundaries. Thus, these findings suggest that TAD boundaries may not behave as absolute barriers to E-P interaction across them.

We found here that a strong TAD boundary may be overcome in vivo, in a cell population-specific manner, in the presence of multiple highly active enhancer clusters or SEs in tandem located in a different TAD than the promoter, and we further demonstrate that their SE-P interactions have strong transcriptional and in vivo functional impact. Intriguingly, HIRE1 and HIRE2 are active, i.e., enriched with H3K27ac, both in *Hoxa2*-expressing PA2 and *Hox*-free Md CNCCs, yet the contacts between HIRE1 and HIRE2 and *Hoxa2* only occur in PA2 CNCCs (Fig. 3a, b). In *Hox*-free CNCCs, the *Hoxa2* promoter may not be available for HIRE1/HIRE2 interaction since, together with the whole *Hoxa* cluster, it is embedded in a large repressive Polycomb domain[16] which may segregate in a repressive nuclear compartment distinct from the active HIRE1/HIRE2. Indeed, Polycomb binding at *Hoxa* promoters in developing limbs can prevent their interaction with active enhancers[97]. In PA2 CNCCs, local patterning signals may activate proximal enhancers so that the *Hoxa2* locus is "singled out" from the Polycomb repressed *Hoxa* cluster and transcriptionally induced, followed by removal of H3K27me3 and switched to H3K27ac deposition[16]; this might, in turn, allow rapid interaction with the inter-TAD SEs boosting *Hoxa2* expression to full transcriptional output. Similarly, *Hoxa3* might become collinearly connected to HIRE1/HIRE2 in PA3 CNCCs, where *Hoxa3* is transcriptionally induced and *Hoxa2* is expressed as well.

Moreover, tissue-specific 3D chromatin conformation can also contribute to enhancer activity and specificity[6]. For instance, the Pen enhancer shows activity in both developing forelimbs and hindlimbs, but it only controls *Pitx1* transcription in hindlimbs. This restricted enhancer activity is associated with a 3D chromatin configuration allowing Pen and *Pitx1* to interact only in hindlimbs, whereas enhancer and promoter are maintained physically separated in forelimbs[6]. Structural chromatin variants can however convert the inactive into an active 3D conformation, thereby inducing *Pitx1* misexpression in forelimbs[6]. Moreover, forced chromatin looping of strong enhancers to developmentally silenced promoters can be sufficient to stimulate transcription[98,99]. Thus, HIRE1/HIRE2-driven transcriptional regulation of *Hoxa2* may be allowed by a PA2-specific 3D chromatin configuration. Furthermore, HIRE1 and HIRE2 might be brought in proximity to the *Hoxa2* locus by a mechanism similar to the domain-skipping interactions described between Drosophila *Scr* and its distal enhancers T1, whereby the formation of an intervening TAD by boundary pairing is essential for distal, inter-TAD, E-P interaction[100]. Active *Hoxa2* and its SE region HIRE1/HIRE2 are both close to interacting TAD boundaries (Fig. 3b), suggesting that pairing between boundary elements might bring distant HIRE1/HIRE2 and its target promoter in proximity by domain-skipping chromatin folding in PA2 CNCCs.

Lastly, HIRE1 and HIRE2 could cooperate with proximal *Hoxa2* enhancers to allow for strong and precise *Hoxa2* expression in the hindbrain and CNCCs. Notably, motif enrichment analysis, and PCHi-C carried out in E10.5 *Hoxa2* full knockout embryos, showed that Hoxa2 itself is partially required to establish long-range interactions in PA2 CNCCs between its own promoter and HIRE2, likely with Pbx and Meis cofactors (Fig. 9). These data support the idea that the *Hoxa2* promoter must be active to recruit HIRE1/HIRE2 and are consistent with

the observation that interconnected autoregulatory loops often exist between SEs and their target promoters[44].

## Methods

### Mice and ethical statement

All animal experiments were approved by the Basel Cantonal Veterinary Authorities under permit number 2670 and conducted in accordance with the Guide for Care and Use of Laboratory Animals.

The generation of *Wnt1::Cre*, *ROSA-tdRFP (Rosa^RFP^)*, and *Hoxa2^EGFP^* were described elsewhere[49,101,102]. The HIRE1 and HIRE2 deletion lines were generated and characterized on the C57Bl/6J background as described in *Mouse genome editing using CRISPR/Cas9* section. All mice were maintained on a mixed background (C57Bl/6J; CD1). The mice were housed in a 12 h light:dark cycle and given ad libitum access to food and water for the duration of the study. The ambient temperature is 22 ± 2 °C and humidity is maintained at 45–65%.

For breeding, one or two female mice were introduced into a cage with a single male and monitored for timed pregnancies. Noon of the day of the vaginal plug was considered as E0.5. All mice used for breedings were at least 8 weeks old and not older than 6 months. To obtain E10.5, E12.5, and E14.5 *Wnt1::Cre;ROSA^RFP^* embryos (denoted as wild-type), we crossed *Wnt1::Cre* transgenic mice[101] with the *Rosa^RFP^* reporter[102] mouse line. To generate E10.5 *Hoxa2^EGFP^* mutant embryos, *Hoxa2^wt/EGFP^* transgenic mice[49] were crossed. To obtain E8.5, E9.5, E10.5, and E18.5 *HIRE1^del/del^* embryos, *HIRE1^wt/del^* mice were crossed. To obtain E18.5 *HIRE1^del/wt^;Hoxa2^EGFP/wt^* fetuses, *Hoxa2^EGFP/wt^* transgenic mice were crossed with *HIRE1^del/wt^* mice. To obtain E18.5 *HIRE2^del/del^* embryos, *HIRE2^del/wt^* mice were crossed. To obtain E18.5 and adult *HIRE2^del/wt^;Hoxa2^EGFP/wt^* specimen, the *Hoxa2^EGFP/wt^* transgenic mouse line and *HIRE2^del/wt^* mice were crossed.

As our study aims to uncover general molecular regulatory mechanisms during facial morphogenesis, sex was not considered in the study design, and findings apply to both sexes. For experiments, cell populations from embryos of different sexes were pooled to carry out molecular analysis, including sequencing.

All mouse lines are available upon request from the corresponding author.

### Mouse genome editing using CRISPR/Cas9

Guide RNAs were designed using CRISPOR[103] to target upstream (Up) and downstream (Dw) the HIRE2 or HIRE1 genomic regions at the following protospacer sequences: HIRE2-Up: AGCACGTAGCACGTC AGTAG; HIRE2-Dw: TGTAGGGTATACTACTAGCC; HIRE1-Up: CACCC AGGAATAGGTGCGTC; HIRE1-Dw: GTGGCCCCCGACTAAACTAT. Deletions of the HIRE2 or HIRE1 genomic regions were performed by using the Alt-R CRISPR-Cas9 system from Integrated DNA Technologies (IDT) and ribonucleoprotein (RNP) complex delivery by mouse zygote electroporation. The RNP complex was composed of two crRNA (Up and Dw the HIRE2 or HIRE1 region), a tracrRNA and the Hifi Cas9 nuclease V3. Following IDT recommendations, a duplex was formed for each guide RNA by mixing an equal volume of crRNA (200 μM) and tracrRNA (200 μM), heated to 95 °C for 5 min, and allowed to cool down at room temperature for 10 min for annealing. The RNP complex was finally prepared by combining and incubating the two crRNA: tracrRNA duplexes (100 μM) and the Hifi Cas9 nuclease V3 (61 μM) in Opti-MEM medium (Gibco), respectively, at the final concentration of 3 μM and 1.2 μM, for 20 min at room temperature. Mouse zygotes (C57Bl/6J) were electroporated with the RNP complex using a NEPA21 electroporator and a CUY501P1-1.5 electrode (Nepa Gene) and re-implanted into foster females. Founder mice harboring the deletion were identified by PCR and sequencing. The mice were genotyped by PCR with the following primers for the HIRE2 deletion (297 bp wildtype and 377 bp deleted fragments): HIRE2-Fw1, 5′-CTTGGTTGGAGGC ATCCTTC-3′; HIRE2-Fw2, 5′-AGGGAGGTTAAAGTATTTAAGTAC-3′; HIRE2-Rv, 5′-CGCAAATTCAGTTCCCAGTAC-3′; and for the HIRE1

deletion (228 bp wildtype and 303 bp deleted fragments): HIRE1-Fw, 5′-ATAGCAGGCACTGAAGCCTG-3′; HIRE1-Rv1, 5′-GTCCTCGGTCCTCATC TCAG-3′; HIRE1-Rv2, 5′-ACCTTAGACAAGGCCTTATCTG-3′.

### Culturing conditions

E14 mESCs, provided by L. Giorgetti's laboratory, were cultured on gelatin-coated culture plates in Glasgow minimum essential medium (Sigma-Aldrich, G5154) supplemented with 15% fetal calf serum (Eurobio Abcys), 1% L-glutamine (Thermo Fisher Scientific, 25030024), 1% sodium pyruvate (Thermo Fisher Scientific, 11360039), 1% MEM non-essential amino acids (Thermo Fisher Scientific, 11140035) 100 μM β-mercaptoethanol, 20 U/ml leukemia inhibitory factor (Miltenyi Biotec, premium grade) in 8% CO2 at 37 °C. The cells underwent mycoplasma contamination testing once a month, and no contamination was found.

### Isolation of cranial neural crest cells (CNCCs)

To collect WT post-migratory CNCCs, we generated E10.5 *Wnt1::Cre;ROSA^RFP^* embryos and micro-dissected the frontonasal process (FNP), the maxillary (Mx) and mandibular (Md) components of the first pharyngeal arch (PA1) and the second pharyngeal arch (PA2). To collect CNCCs of the developing pinna, we generated E12.5 and E14.5 *Wnt1::Cre;ROSA^RFP^* embryos and micro-dissected the forming pinna prominence. The *Wnt1* promoter drives Cre expression in CNCC pre-migratory progenitors, resulting in permanent RFP reporter activity in their post-migratory progeny[101]. Fluorescence-activated cell sorting (FACS) was used to isolate the RFP-positive cranial CNCCs. Further processing of these cells was dependent on the downstream application (e.g., RNA-seq, ATAC-seq, ChIP-seq, Hi-C, or promoter capture Hi-C)—see below.

For the *Hoxa2* loss of function analysis, Md and PA2 from E10.5 *Hoxa2^EGFP/EGFP^* mutant embryos were micro-dissected and the GFP-positive CNCCs were isolated by FACS. The processing of these cells was adapted for promoter capture Hi-C.

To collect cells from *HIRE1^del^* and *HIRE2^del^*, we generated E10.5 embryos and micro-dissected PA2 and PA3. The processing of the cells was adapted for RNA-seq.

### Sample preparation, RNA isolation and sequencing (RNA-seq)

Dissected tissue from E10.5 and E12.5 embryos was kept in 1× PBS on ice, then treated with 0.5% trypsin/1× EDTA at 37 °C for 10 min and immediately put on ice. Dissected tissue from E14.5 embryos was kept in 1× PBS on ice, then treated with papain digestion mix (10 mg ml⁻¹ papain, 2.5 mM cysteine, 10 mM HEPES (pH 7.4), 0.5 mM EDTA, and 0.9× DMEM) for 7 min at 37 °C and immediately put on ice. The tissue was rinsed once in ice-cold 1× DMEM/10% FBS, followed by two washes in ice-cold 1× DMEM. The tissue was dissociated by pipetting. CNCCs from embryos with genotype *Wnt1::Cre;ROSA^RFP^* were filtered and collected by FACS. After sorting, CNCCs were pelleted by centrifugation for 5 min, 200×*g* at 4 °C. Total RNA was extracted using the Single Cell RNA Purification Kit (NORGEN, 51800) with genomic DNA digestion using an RNase-Free DNase I Kit (NORGEN, 25710) according to the manufacturer's protocol. For each sample, three independent biological replicates were prepared.

For sequencing of total RNA, unstranded RNA-Seq libraries were prepared using Illumina TruSeq RNA Library Prep Kit v2 according to manufacturers' instructions. Sequencing was performed on an Illumina HiSeq 2500 machine (50 bp read length, single-end) or an Illumina NovaSeq 6000 machine (50 bp read length, paired-end).

### Sample preparation, chromatin immunoprecipitation and sequencing (ChIP-seq)

Tissue from E10.5, E12.5, and E14.5 embryos was dissociated as described for RNA-seq. Then the dissociated cells were cross-linked with 1% formaldehyde for 10 min at room temperature (RT) and

quenched with glycine (final concentration 125 mM) for 5 min at RT. Cells were spun down (500×*g*, 10 min, 4 °C). CNCCs with genotype *Wnt1::Cre;ROSA*<sup>RFP</sup> were filtered and collected by FACS.

ChIP experiments with anti-H3K4me2 (Millipore, Cat. #07-030), anti-H3K27me3 (Millipore, Cat. # 07-449) anti-H3K27ac (Abcam, ab4729), or anti-CTCF (Cell Signaling, CST 2899) antibodies were performed as described in ref. 16.

ChIP libraries were generated using bar-coded adapters (NEB, E7335) in combination with the NEBNext® Ultra™ DNA Library Prep Kit for Illumina® (NEB, E7370) according to manufacturers' instructions. The quality of the libraries and size distribution was assessed on an Agilent 2100 Bioanalyzer (Agilent Technologies). All ChIP-seq experiments were performed with two independent biological replicates, except for CTCF, only one biological replicate was prepared per sample. Sequencing was performed on an Illumina HiSeq 2500 machine (50 bp read length, single-end).

### Sample preparation, assay for transposase-accessible chromatin and sequencing (ATAC-seq)

Tissue from E10.5, E12.5, and E14.5 embryos was dissociated as described for RNA-seq. CNCCs with genotype *Wnt1::Cre;ROSA*<sup>RFP</sup> were filtered and collected by FACS.

To identify open chromatin regions, we used the Assay for Transposase-Accessible Chromatin (ATAC-seq) protocol, performed according to ref. 104. with minor modifications as described previously in ref. 16. For each sample, at least two independent biological replicates were prepared.

Sequencing was performed on an Illumina NextSeq 500 (75 bp read length, paired-end).

### Hi-C and PCHi-C sample preparation and sequencing

Tissue from E10.5, E12.5, and E14.5 embryos was dissociated as described for RNA-seq. E14 mESCs were collected with Accutase (Sigma-Aldrich, A6964) and resuspended in cold 1× PBS. Then the dissociated cells were cross-linked with 2% paraformaldehyde for 10 min at room temperature (RT) and quenched with glycine (final concentration 250 mM) for 5 min at RT. Cells were spun down (500×*g*, 10 min, 4 °C) and resuspended in cold 1× PBS. CNCCs with genotype *Wnt1::Cre;ROSA*<sup>RFP</sup> or *Hoxa2*<sup>EGFP/EGFP</sup> were filtered and collected by FACS. Cells were spun down (500×*g*, 10 min, 4 °C), the supernatant was removed, and cell pellets were flash-frozen and stored at −80 °C.

Promoter-capture Hi-C was performed as described previously in ref. 105 with adaptations to our system. About 39021 biotinylated RNA probes were designed to enrich 22,225 annotated gene promoters in the mouse genome using PCHi-C. These probes were designed and used as described in refs. 37, 105 and purchased from Agilent Technologies. The input material per PCHi-C experiment was 2.5–5 × 10⁶ cells and per Hi-C experiment, 1–2 × 10⁶ cells. Cells were incubated in 1 mL (Hi-C: 500 μL) ice-cold lysis buffer (10 mM Tris-HCl pH 8, 0.2% (vol/vol) NP-40, 10 mM NaCl, 1× protease inhibitor cocktail (Roche)) for 30 min with periodical mixing, followed by centrifugation to pellet nuclei (760×*g*, 5 min, 4 °C). Cell nuclei were resuspended in 358 μL (Hi-C: 179 μL) ice-cold 1.25× NEBuffer 2, 11 μL (Hi-C: 5.5 μL) of 10% SDS was added, and nuclei were incubated at 37 °C, shaking at 950 rpm, for 1 h. Then 75 μL (Hi-C: 37.5 μL) of 10% Triton X-100 was added and cell nuclei were incubated at 37 °C, shaking at 950 rpm, for 1 h. A 5 μL aliquot was taken as a control for undigested chromatin (stored at −20 °C). For HindIII digestion, 10 μL (Hi-C: 5 μL) of HindIII enzyme (NEB, cat. no. R0104T) was added, incubated 6 h at 37 °C shaking at 950 rpm, followed by the addition of another 10 μL (Hi-C: 5 μL) of HindIII and overnight incubation at 37 °C shaking at 950 rpm. A 5 μL aliquot was taken to test digestion efficiency (stored at −20 °C). To fill-in and biotinylate the restriction fragment overhangs 60 μL (Hi-C: 30 μL) of biotinylation master mix (6 μL 10× NEBuffer 2, 2 μL H₂O, 1.5 μL 10 mM dATP, 1.5 μL 10 mM dGTP, 1.5 μL 10 mM dTTP, 37.5 μL 0.4 mM biotin-14-

dATP, and 10 μL 5 U/μl Klenow (DNA polymerase I large fragment, NEB M0210L)) was added, incubated for 75 min at 37 °C mixing regularly by pipetting, and then placed on ice. Next, 454 μL (Hi-C: 227 μL) ligation master mix (100 μL 10× ligation buffer (NEB B0202), 10 μL 10 mg/mL BSA, 344 μL nuclease-free H₂O) and 25 μL (Hi-C: 12.5 μL) 1 U/μL T4 DNA ligase (Invitrogen 15224-024) was added for overnight blunt end ligation in a water bath at 16 °C. The next day ligated samples were incubated for a further 30 min at room temperature. The re-ligated chromatin products and test aliquots were de-cross-linked for at least 6 h by adding 100 μL (Hi-C: 50 μL) and 2.5 μL proteinase K (10 mg/mL), incubated at 65 °C. Afterward, 12 μL (Hi-C: 6 μL) or 0.5 μL of 10 mg/ml RNase A was added to the samples and test aliquots, respectively, and incubated for 60 min at 37 °C. Next, chromatin was precipitated by adding 1 volume of phenol:chloroform:isoamyl alcohol (25:24:1) (PC) to the samples and test aliquots, vigorously shaking them, followed by centrifugation at a maximum speed of a tabletop centrifuge (-16,500×*g*) at room temperature for 5 min. The upper phase containing the chromatin was transferred to a new 15 mL tube. To reduce loss of DNA, 1 mL of TLE buffer (10 mM Tris-HCl (pH 8.0), 0.1 mM EDTA) was added to the tube containing PC and residual chromatin, tube was vigorously mixed, followed by centrifugation at a maximum speed of a tabletop centrifuge (-16,500×*g*) at room temperature for 5 min. The upper phase containing the chromatin was transferred to the 15 mL tube. One-tenth volume of 3 M sodium acetate was added to the chromatin sample mixed by vortexing, followed by the addition of 3 volumes of 100% ethanol and vortexing to mix. The samples were frozen at −80 °C for at least 2 h. The precipitated chromatin was isolated by centrifugation at maximum speed for 45 min at 4 °C. The chromatin pellet was washed twice with 2 mL freshly prepared 70% ethanol and centrifuged at maximum speed for 15 min at 4 °C in between washes. Finally, the chromatin pellet was dried at room temperature and resuspended in 100 μL TLE buffer. To check the quality of the Hi-C library DNA, 1:50 dilution of the sample, and the undigested and digested controls were loaded on a 1.5% agarose gel. The DNA concentration of the Hi-C library was determined using Qubit™ dsDNA HS assay kit.

For Hi-C, 5 μg DNA and for PCHi-C, 10 μg DNA (split into two 5 μg aliquots) were used for library preparation. The next steps were performed per 5 μg aliquot. The DNA was sonicated using Covaris S220 (peak incidence power: 175 W; duty factor: 10%; cycles per burst: 200; time: 120 s) in a volume of 130 μL in a Covaris microTUBE to obtain 100–1000 bp long DNA fragments. End repair, dATP-tailing, and Adapter ligation was performed on beads. For this, 100 μL of Dynabeads™ MyOne™ Streptavidin T1 (Invitrogen) were washed twice with 400 μL Tween Buffer (TWB; 5 mM Tris-HCl pH 8.0, 0.5 mM EDTA, 1 M NaCl, 0.05% Tween 20) and resuspended in 400 μL 2× binding buffer (2×BB; 10 mM Tris-HCl pH 8.0, 1 mM EDTA, 2 M NaCl). The volume of the sonicated DNA was increased to 400 μL with TLE. DNA was mixed with washed beads (800 μL total volume) and incubated at RT for 45 min rotating at 5 rpm. Beads were reclaimed on a magnetic separation stand, the supernatant was removed, followed by a wash with 400 μL 1×BB, then with 100 μL 1×ligation buffer, and resuspended in 50 μL 1× ligation buffer. For end repair, each aliquot was mixed on ice with 50 μL 2.5 mM dNTP mix (12.5 μL of 10 mM of each dNTP), 18 μL of T4 DNA Polymerase (NEB M0203), 18 μL of T4 PNK (NEB M0201), 3.7 μL of DNA polymerase I large fragment (Klenow; NEB M0210), and 360.1 μL of H₂O, and then incubated at 20 °C for 60 min. Beads were reclaimed on a magnetic separation stand, and the supernatant was removed, followed by two washes with 500 μL 1×TWB, one wash with 500 μL 1×BB, one wash with 500 μL TLE and resuspended in 415 μL TLE. For dATP-tailing, each aliquot was mixed with 50 μL 10× NEBuffer 2, 5 μL 10 mM dATG, and 20 μL 5 U/μL Klenow Fragment (3′ → 5′ exo-) (NEB, M0212), and then incubated at 37 °C for 30 min. Beads were reclaimed on a magnetic separation stand, the supernatant was removed, followed by two washes with 500 μL 1×TWB, one wash with

500 μL 1×BB, one wash with 200 μL 1× ligation buffer, and resuspended in 100 μL 1× ligation buffer. For PCHi-C adapter ligation, 4 μL 15 μM pre-annealed PE adapters (described in ref. 105.) and 4 μL NEB T4 Ligase 400 U/mL (NEB, M0202) were added to each aliquot and incubated 2 h at room temperature. For Hi-C, 10 μL 15 μM TruSeq DNA Sgl index adapters (Illumina, Cat. No. 20016329) and 4 μL NEB T4 Ligase 400 U/mL (NEB, M0202) were added to each aliquot and incubated for 2 h at room temperature. Beads were reclaimed on a magnetic separation stand, the supernatant was removed, followed by two washes with 500 μL 1×TWB, one wash with 500 μL 1× BB, one wash with 200 μL 1×BB, one wash with 100 μL 1× NEBuffer 2, one wash with 50 μL 1× NEBuffer 2 and resuspended in 50 μL 1× NEBuffer 2. At this point, all aliquots per PCHi-C sample were pooled.

For on-bead PCHi-C and Hi-C library amplification, a reaction master mix was prepared, reaction volume per PCR tube was 25 μL (PCHi-C: 5 μL 5× Herculase II reaction buffer (Agilent), 0.25 μL 100 mM dNTPs, 2 μL PE PCR primer 1.0[105], 2 μL PE PCR primer 2.0[105], 0.5 μL Polymerase (Herculase, Agilent), 2.5 μL Hi-C DNA on beads, 12.75 μL nuclease-free $H_2O$; Hi-C: 12.5 μL KAPA HiFi HotStart ReadyMix (Roche), 2.5 μL Illumina primer mix (contained in KAPA HiFi HotStart Library Amplification Kit), 2.5 μL Hi-C DNA on beads, 7.5 μL nuclease-free $H_2O$) and PCR was run under the following conditions: 30 s at 98 °C; 6–7 cycles of 10 s at 98 °C, 30 s at 65 °C, 30 s at 72 °C; 7 min at 72 °C. Post-amplification PCR reactions per sample were pooled and DNA was cleaned up using AMPure XP beads and eluted in nuclease-free $H_2O$. Obtained DNA was used directly for Hi-C sequencing or for promoter-capture Hi-C as described in ref. 105. Hi-C experiments on mESC and cells from E12.5 and E14.5 pinna were performed in biologically independent duplicates. Hi-C experiments on Md and PA2 at E10.5 have one replicate each. All promoter-capture Hi-C experiments have two biologically independent replicates. Sequencing was performed on an Illumina NextSeq 500 (75 bp read length, paired-end).

## Quantitative real-time PCR (qRT-PCR)

Total RNA was extracted from PA2 of E10.5 embryos using the Single Cell RNA Purification Kit (NORGEN, 51800) with genomic DNA digestion using an RNase-Free DNase I Kit (NORGEN, 25710) according to the manufacturer's protocol. cDNA synthesis was performed using Superscript III (Invitrogen, ref#18080-044) according to the manufacturer's instructions and was then treated with RNAse H (NEB, M0297L) for half an hour at 37 °C. Samples were diluted 1:10 and qPCR was performed using StepOnePlus real-time PCR machine (Thermo Fisher) with SYBR Green PCR master mix (Thermo Fisher, ref#4309155) according to manufacturer's protocol.

The following primers were used: *Hoxa2* forward primer (FW) 'CAAGACCTCGACGC TTTCACAC', reverse primer (RV) 'CCTTCATCC AGGGATACTCAGGC'; *Gapdh* FW 'GAGAGGCCCTATCCCAACTC', RV 'GGTCTGGGATGGAAATTGTG'. The *Hoxa2* primers were specifically designed to only amplify from the *Hoxa2* wild-type allele but not the *Hoxa2* EGFP allele. Primer efficiencies of each primer pair were evaluated with a standard curve, and the occurrence of primer dimers was checked for with a melting curve and gel electrophoresis. Relative expression levels of *Hoxa2* were quantified using the ΔΔCt method, where *Hoxa2* Ct values were normalized to *Gapdh* levels and the average ΔCt of the wild-type samples. Statistical analysis was performed by one-way analysis of variance (ANOVA) followed by Dunnett's test.

## Skeletal staining

E18.5 mouse embryos were skinned and eviscerated. Skeletal staining of the embryos was performed according to a previously described protocol[74]. Samples were fixed in 95% ethanol for 5–7 days. Subsequently, embryos were incubated with 0.015% alcian blue 8GS, 0.005% alizarin red S, and 5% acetic acid for 3 days with agitation at 37 °C. Samples were cleared in 1% KOH for several days and in 1% KOH/

glycerol series until surrounding tissues turned transparent. The samples were stored in glycerol for a longer term.

## Whole-mount in situ hybridization (WISH)

In situ hybridization on whole-mount embryos was performed as previously described in ref. 106 with minor modifications. Embryos were dissected free of extraembryonic membranes in 1× PBS containing 0.1% Tween 20 (PBT) and fixed in 4% paraformaldehyde in 1× PBS for 2 h at room temperature or at 4 °C overnight. Throughout the procedure, the embryos were rocked gently on a mechanical rocking platform unless otherwise indicated. Embryos were washed three times with PBT, dehydrated through a methanol-PBT series into 100% methanol, and stored at −20 °C until further use. To summarize, embryos were rehydrated through a methanol-PBT series and washed three times in PBT, followed by a 1-h incubation in 6% hydrogen peroxide/PBT at room temperature and three washes in PBT. Next, embryos were treated with 5–10 μg/ml proteinase K in PBT for 1–2 min at room temperature (Proteinase K concentration and incubation time must be adjusted according to embryo size and a batch of Proteinase K), directly followed by refixation in 0.2% glutaraldehyde and 4% paraformaldehyde in 1× PBS for 20 min at room temperature and three washes with PBT. Embryos were pre-hybridized for 3–4 h at 70 °C in hybridization buffer (50% deionized formamide, 5× saline-sodium citrate (SSC) buffer pH 7.5, 100 μg/ml tRNA, 50 μg/ml heparin, 1% SDS). The hybridization buffer was replaced, RNA probes labeled with digoxigenin were added, and the embryos were incubated overnight at 70 °C. The embryos were rinsed once with Wash 1 (50% formamide, 4× SSC pH 7.5, 1% SDS), followed by two washes for 30 min each at 70 °C in Wash 1. The embryos were washed once at 70 °C for 10 min in a 1:1 mix of Wash 1 and Wash 2 (0.5 M NaCl, 10 mM Tris-HCl pH 7.5, 0.1% Tween 20), then three times in Wash 2 for 5 min at room temperature. The wash buffer was replaced by Wash 2 containing 100 μg/ml RNase A and incubated at 37 °C for 1 h. The embryos were washed once for 5 min with Wash 2, then once for 5 min at 70 °C with pre-heated Wash 3 (50% formamide, 2× SSC pH 7.5, 0.1% SDS). This was followed by two washes with Wash 3 at 70 °C for 1 h each. The embryos were washed three times with 1× maleic acid buffer containing 0.1% Tween 20 (MABT), then incubated for 2–3 h in MABT with 2% blocking reagent (Roche-11096176001). Sufficient embryo powder was heat-inactivated just before use in MABT with 2% blocking reagent 70 °C for 30 min, vortexed for 5 min, and placed on ice. Then, the required amount of anti-DIG AP FAB fragment (Roche-11093274910) was preabsorbed for 1 h at 4 °C with prepared embryo powder in MABT with 2% blocking reagent, followed by centrifugation at 4 °C for 10 min. The supernatant was diluted in MABT with 2% blocking reagent to obtain a final antibody dilution of 1:2000. Embryos were incubated with the preabsorbed antibody overnight at 4 °C. The embryos were washed three times at room temperature with MABT containing 2 mM levamisole, followed by eight washes for 1 h each at room temperature. The last wash was followed by three changes of NTMT (100 mM NaCl, 100 mM Tris-HCl pH 9.5, 50 mM MgCl2, 0.1% Tween 20) containing 2 mM levamisole. The color reaction was initiated by placing the embryos into NTMT containing 2 mM levamisole, 3.5 μL/mL NBT (Roche-11383213001), and 3.5 μL/mL BCIP (Roche-11383221001). Staining was allowed to proceed for multiple hours at room temperature in the dark. When staining was satisfactory, the embryos were washed three times with MABT, and left to wash in MABT at 4 °C until the background signal was sufficiently removed.

The following RNA probes were used: *Hoxa1*[107], *Hoxa2*[108], and *Hoxa3*[23,24].

## Histological analysis and in situ hybridizations

Mouse embryos were collected and fixed in 4% PFA/1× PBS overnight at 4 °C. For cryostat sections, tissues were cryoprotected in 20% sucrose/1× PBS and embedded in 7.5% gelatin/10% sucrose/1× PBS before being

frozen at −80 °C. Cryostat sections (25 μm) were cut (CryoStar NX70) in sagittal orientation.

Frozen sections were fixed in 4% PFA/1× PBS, and washed in PBS. Acetylation was performed with acetic anhydride solution (1.35% (v/v) Triethanolamine, 0.175% (v/v) HCl, 0.25% (v/v) acetic anhydride) for 10 min at RT, before being pre-hybridized with Hybridization buffer (50% deionized formamide, 1× Salt, 10% dextran sulfate, 1 mg/ml tRNA, 1× Denhardt) for 1 h 30 at room temperature. Hybridization was performed with antisense riboprobes labeled with digoxigenin-11-d-UTP diluted 1:100 in hybridization buffer, for 16–18 h at 70 °C, in a humid chamber (humidified with 50% formamide/5× SSC). On day 2, Sections were rinsed with 5× SSC pH 4.5, followed by washing 2 h at 70 °C in 0.2× SSC pH 4.5, and 2× 10 min at RT in MABT buffer (0.1 M Maleic Acid, 0.15 M NaCl, 2 N NaOH, pH 7.5). After blocking 1h30 at RT in MABT/2% Blocking reagent (Roche Diagnostics), sections were incubated with anti-DIG antibody conjugated to alkaline phosphatase in MABT/2% Blocking reagent (1:5,000; Roche Diagnostics) overnight at RT. On day 3, slides were washed 8x In MABT (0.1 M Maleic Acid, 0.15 M NaCl, 2 N NaOH, pH 7.5), 2× 10 min in NTMT buffer (100 mM Tris-HCl, pH 9.5, 100 mM NaCl, 50 mM MgCl$_2$, 0.1% Tween 20, pH 9.5). The alkaline phosphatase activity was detected using NBT and BCIP diluted in NTMT buffer at 4 °C. To stop the reaction, slides were washed 2× 10 min in MABT. Slides were mounted with aqueous mounting media (Aquatex, Merck). Imaging was performed using Zeiss Axio Scan Z1, using 5× (NA 0.25), 10× (NA 0.45), and Zeiss upright Axio imager Z1 using 5× (NA 0.25).

The following RNA probes were used: *Hoxa2*[108] and *Hoxa3*[23,24].

## Use of previously published datasets

Some datasets have been used from our previously published work[16], as indicated in the manuscript, and are published in the SuperSeries GSE89437. A number of RNA-seq, ChIP-seq, and ATAC-seq samples published in this series have been re-processed for this study (Supplementary Data 7).

## Computational analysis

**Reference genome and annotation.** The mouse GRCm38/mm10 genome assembly was used as a reference. Gene annotation was obtained from the *TxDb.Mmusculus.UCSC.mm10.knownGene* Bioconductor package (https://doi.org/doi:10.18129/B9.bioc.TxDb.Mmusculus.UCSC.mm10.knownGene, version 3.4.7). For RNA-seq analysis, a single transcript start site (TSS) was selected randomly for each gene, and promoter regions were defined as a 2 kb window centered on the TSS.

For genome browser views, the number of alignments per 100 bp window in the genome and per million alignments in each sample were calculated, stored in BigWig format using the *qExportWig* function from the *QuasR* Bioconductor package (version 1.34.0)[109] and visualized as custom tracks in either UCSC genome browser (http://genome.ucsc.edu)[110] or pyGenomeTracks (version 3.7)[111].

## RNA-seq data analysis

All RNA-seq experiments used in this study were sequenced single-end with 50 bp read length or paired-end 50 bp read length (Supplementary Data 7). The dataset containing single-end reads were not compared with the datasets containing paired-end reads. Thus, trimming the second read of the paired-end data was not necessary. Reads of single-end and paired-end datasets were aligned to the reference genome using the *QuasR* Bioconductor package (version 1.34.0)[109] by the *qAlign* function with parameters *aligner = "Rhisat2"* (version 1.12.0) and *splicedAlignment = TRUE*. Alignments overlapping genes from the *TxDb.Mmusculus.UCSC.mm10.knownGene* Bioconductor package (https://doi.org/doi:10.18129/B9.bioc.TxDb.Mmusculus.UCSC.mm10.knownGene, version 3.4.7) were quantified using the *qCount* function with parameter *orientation = "opposite"*.

The RNA-seq samples for E10.5 Md and E10.5 PA2 from public datasets (Supplementary Data 7) were processed in the same way for illustration as genome browser tracks.

## Differential RNA-seq analysis

To identify differentially expressed genes, only expressed genes in both biological replicates are used (CPM > = 1). Differentially expressed genes are identified using *edgeR* (version 3.38.1)[112]. Two different models are fit for genes counts:

1/ ~ time point (E10.5 PA2; E12.5 Pinna; E14.5 Pinna)

2/ ~ genotype (PA2 WT and PA2 delta E3; PA3 WT and PA3 delta E3)

For each model, dispersions are estimated using *estimateDisp* and statistical significance is calculated using *glmQLFTest*. Differentially expressed genes are defined as genes with an FDR less than 0.05.

## ChIP-seq data analysis

Reads were mapped to the reference genome using *bowtie2 w*ith default settings (version 2.4.2)[113,114] and converted to bam files using *samtools* (version 1.2)[115,116].

## ATAC-seq data analysis

The adapter sequence CTGTCTCTTATACACA was trimmed from the 3' end of all samples using *cutadapt* (version 3.7)[117] with overlap = 1. The trimmed reads were aligned with *bowtie2* (version 2.4.2) with the options *–fr, –minins 0, –maxins 1000, –nodiscordant* and *–dovetail*, and converted to bam files using *samtools* (version 1.2)[115,116].

## Hi-C data analysis

The Hi-C data has been mapped and quality controlled with *HiCUP* (version 0.6.1)[118], the interaction matrix was created with *Juicer* (version 1.6)[119], and was transformed into a *cool* matrix[120] with *hic2cool* (https://github.com/4dn-dcic/hic2cool, version 0.8.3) and *HiCExplorer's hicConvertFormat* (version 3.7.2)[121]. For the Hi-C on mESC, E12.5 pinna, and E14.5 pinna the *cool* matrices of biological replicates were merged into one *cool* matrix with *HiCExplorer's hicSumMatrices* (version 3.7.2)[122]. The merged *cool* matrix was normalized with *HiCExplorer's hicCorrectMatrix* (version 3.7.2)[122] using KR normalization. The TAD calling was applied by *HiCExplorer's hicFindTADs* (version 3.7.2)[122,123]. The Hi-C and TAD data was plotted with *pyGenomeTracks* (version 3.7)[111].

The differential interaction maps were generated by using the *FAN-C* suite of tools (Kruse et al., 2020). To derive the differential interactions maps, the normalized scores of one matrix were subtracted from the other using f*anc compare* tool, and the maps are plotted using *fancplot* tool.

Virtual 4C plots are generated by using *Virtual4CPlot* tool from *FAN-C*.

## PCHi-C data analysis

Capture HiC libraries are first analyzed with *HiCUP* (version 0.6.1)[118], then significant interactions are identified using *CHiCAGO* (version 1.24.0)[124]. *CHiCAGO* pipeline is used with the recommended parameters for six cutter restriction enzymes[125]: *minFragLen = 150* and *maxFragLen = 40000, maxLBrownEst = 1500000* and *binsize = 20000*.

Interactions with a *CHiCAGO* score of 5 are considered as high-confidence interactions. Significant interactions are plotted as arcs linking baits to OE using the python api pygenometrack.

4C-like profiles of interactions are plotted using ChiCMaxima (version 1.0) browser[126]. The PCHi-C CHiCAGO interactions with a score ≥5 were plotted with *pyGenomeTracks* (version 3.7)[111].

The percentage of decrease of interactions of Hoxa2 promoter and HIRE2 SE in the PA2 E10.5 *Hoxa2*$^{EGFP/EGFP}$ compared to PA2 E10.5 WT is calculated using quantile normalized raw counts of Hoxa2 interactions with the HIRE2 region.

## CTCF motif occurrences

To identify the orientation of CTCF motifs, we used the position frequency matrix of CTCF for Mouse downloaded from Jaspar 2022[127]. CTCF motif occurrences are identified by the *FIMO* package (version 5.4.1)[128]. Only motif sites with p-values <= 1.0E-4 are included. In total, 380951 CTCF motif occurrences were identified.

## SE calling

SEs were identified computationally by referring to the ROSE algorithm[44,46]. First, bam files of the H3K27ac ChIP-seq alignments from all E10.5 samples (Supplementary Data 7) were merged. Then, peaks were called using MACS2 (version 2.1.3.3)[129] with the options *–nomodel,–shift 0,–extsize 141,–keep-dup all,–qvalue 0.001,–cutoff-analysis*. The extension size had been determined previously based on cross-correlation using the *csaw* Bioconductor package (version 1.28.0)[130]. Peaks that overlapped with promoter regions, i.e., 2500 bp upstream or downstream of the transcription start sites of the UCSC known genes (*TxDb.Mmusculus.UCSC.mm10.knownGene*, v. 3.10.0), or peaks overlapping with blacklisted regions[131] were removed. Only peaks located on chromosomes 1–19 or the X-chromosome were kept. Counts per million (CPM) were quantified for all remaining peaks for each sample individually using *QuasRs qCount* function with default parameters[109]. The quantified peaks were merged into larger regions if they were less than 12.5 kb apart. Subsequently, the CPMs within the individual peaks were summed. If the resulting stitched regions overlapped with the promoter regions, they were removed. The remaining regions were ranked by their summed counts for each sample. The threshold to distinguish between super- vs. non-SEs was determined as in ROSE. Lastly, only regions with at least 3 individual peaks were kept to select clusters of enhancers.

## Calculation of interaction strengths between SEs and promoters

SEs were matched to differentially expressed transcription factor encoding genes if at least one significant interaction (Chicago score ≥5) was detected from the promoter bait to a restriction fragment overlapping with a SE region in at least one population. To ensure the interaction occurs in a population where both elements are active, the SE had to show acetylation levels above the relative threshold, and the gene had to be expressed with >2 RPKM in the same population.

To obtain the interaction strengths between SEs and associated transcription factor promoters, we quantified the mean Chicago score for interactions between all restriction fragments overlapping with a SE and the corresponding promoter for each sample.

To display the association between SE acetylation, link strength, and gene expression, a heatmap was generated with the *ComplexHeatmap* Bioconductor package (version 2.10.0)[132]. Rows were clustered using k-means clustering on the relative gene expression levels with seven centers.

## Motif enrichment analysis

For transcription factor motif analysis, ATAC-seq data from E10.5 PA2 and E12.5 Pinna was used. Peaks were identified separately for each dataset using MACS2 (version 2.1.3.3)[129] with options *–nomodel,–shift −100,–extsize 200, -f BED,–gsize 1.87e9*, and *–qvalue 0.10* and a combined peak set was created by fusing peaks from E10.5 PA2 and E12.5 Pinna (*n* = 133,466). Since each ATAC sample was a mixture of mouse embryos of different sexes, only autosomal peaks were used for the analysis. Peaks were classified as "promoter" peaks, when the distance of a peak midpoint to the nearest TSS was smaller than 1 kb, or else as "non-promoter" peaks (hereon called enhancers). The ATAC-seq signal of E10.5 PA2 and E12.5 Pinna replicates was quantified individually at autosomal enhancers (*n* = 106,587) using *QuasR's qCount* function with default parameters[109] and normalized by dividing by the total number of alignments in each sample and multiplying by 1e6 to obtain counts

per million (CPM). For further analysis, the CPM values were log2-transformed with a pseudo-count of 1. Enhancers overlapping the genomic interval between HIRE2 and HIRE1 (chr6:50789172-51087888, mouse GRCm38/mm10) were extracted (*n* = 31). The average logCPM at each extracted enhancer was calculated and subtracted from the individual logCPM values of each sample to obtain the relative accessibility per enhancer and per sample. According to the relative accessibility values for each sample, the extracted enhancers were grouped into three clusters and heatmaps were drawn using the *ComplexHeatmap* Bioconductor package (version 2.6.2)[132]. The mean accessibility profile for each cluster was calculated and used to rank all enhancers by their similarity to each cluster profile (visualized using *vioplot* function from the *vioplot* package version 0.3.6). The top 1000 enhancers with the highest similarity in their accessibility profile compared to each cluster profile were grouped together (visualized using *upset* function from the *UpSetR* package version 1.4.0) and all residual peaks were grouped together as a control set. The *calcBinnedMotifEnr* function from the *monaLisa* package (version 0.1.40)[62] with *method = "R"* was used to identify what motifs are enriched in each of the three cluster profiles using the vertebrate list of motifs present in JASPAR (JASPAR2018)[133]. Motifs that had an FDR less than 0.001 in any of the three cluster profiles were selected. Heatmaps were drawn using the *ComplexHeatmap* Bioconductor package. JASPAR motifs were extracted from *JASPAR2018* package (version 1.1.1) using *getMatrixSet* function from the *TFBSTools* package (version 1.28.0) with *opts = list(tax_group = "vertebrates")*.

## Selection of representative motifs

All motifs that had an FDR less than 0.001 in any of the three cluster profiles from the motif enrichment analysis were used to scan the sequences of the 31 putative enhancers in the region from HIRE2 to HIRE1 for motif hits using the *findMotifs* function from the *monaLisa* package (version 1.4)[62] with *min.score = 10.0* and *method = "matchPWM"*. Since the enhancers/ATAC peaks differ in lengths, the number of hits per peak per kb were calculated. For each motif, the average motif rate per kb across all enhancers was determined. To further summarize these motifs and select the most relevant ones, motifs were clustered based on the average motif hit rate per kb into two sets using *k-means* clustering. All the motifs in the cluster with the higher hit rate were used to perform a motif similarity analysis using the *motifSimilarity* function from the *monaLisa* package (version 1.4)[62] with *method = "R"*, which compares all pairs of motifs. The results of this analysis were grouped into ten clusters using *k-means* clustering (k = 10) and a heatmap was drawn using the *ComplexHeatmap* Bioconductor package (version 2.14.0)[132]. For each cluster, the motif with the least average distance to the other motifs in the cluster was selected as a representative motif.

## Statistics and reproducibility

No statistical method was used to predetermine the sample size. No data were excluded from the analyses. The experiments were not randomized. The Investigators were not blinded to allocation during experiments and outcome assessment.

## Reporting summary

Further information on research design is available in the Nature Portfolio Reporting Summary linked to this article.

## Data availability

All raw sequencing data and processed data generated in this study have been deposited and are publicly available in the Gene Expression Omnibus (GEO) under GEO Series accession number GSE211904 (all data SuperSeries), GSE211899 (ATAC-seq), GSE211900(ChIP-seq), GSE211901 (Hi-C), GSE211902 (PCHi-C), and GSE211903 (RNA-seq). The following public sequencing datasets were published in ref. 16. and

available through GEO were used in this study: ChIP-seq, RNA-seq, and ATAC-seq data from mouse E10.5 cranial neural crest cell subpopulations (GSE89437). FACS gating strategies are presented in Supplementary Fig. 3. ChIP-seq peaks for Hoxa2 in PA2 at E11.5 were obtained from ref. 57, and ChIP-seq peak data for Pbx and Meis in PA2 at E11.5 were obtained from ref. 58. Further public databases used in this study are UCSC (mm10 reference genome assembly, gene annotation), JASPAR2018 (vertebrate transcription factor motifs)[133], JASPAR 2022 (vertebrate transcription factor motifs)[127], and the ENCODE Blacklist[131].

## Code availability

Computational analyses were performed in R and Python using the publicly available packages described in Methods and Reporting Summary.

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

## Acknowledgements

We thank S. Smallwood, François Spetz, and FMI facilities for excellent technical support and T. Kitazawa, D. Machlab, and members of the Rijli group for discussion. We thank T. Sexton for valuable input on capture HiC technology. We thank L. Giorgetti for the E14 mESCs line. O.J. was supported by an EMBO Long-Term fellowship. F.M.R. was supported by the Swiss National Science Foundation (CRSII5_173868 and 31003A_175776) and the Novartis Research Foundation. This project has also received funding from the European Research Council (ERC) under the European Union's Horizon 2020 research and innovation program (grant agreement No 810111-EpiCrest2Reg).

## Author contributions

S.K., M.M., and F.M.R. conceived the study, designed experiments, and analyzed experimental data. S.K. performed most of the experiments. H.K. carried out cell sorting. S.K. and M.B.S. performed motif enrichment analysis. S.K. and J.W. performed alignments and QC of ChIP-seq, RNA-seq, and ATAC-seq data. F.R. performed a super-enhancer analysis. Y.B.Z. performed alignments and QC of Hi-C and promoter-capture Hi-C data and carried out differential gene expression analysis of RNA-seq and analysis of Hi-C and promoter-capture Hi-C data. J.W. analysed Hi-C data from mESC and generated multiple plots and TAD separation scores. O.J. performed promoter-capture Hi-C. S.D. generated CRISPR/Cas9 knockout mice. N.V. carried out in situ hybridizations on sections. A.S. carried out Hi-C in mESCs. M.M., S.K., and F.M.R. analyzed HIRE1 and HIRE2 mutant phenotypes. M.M. and S.K. wrote the first draft, further revised by O.J. and Y.B.Z. F.M.R. revised and wrote the final manuscript. All authors commented on the manuscript.

## Competing interests

The authors declare no competing interests.
