## [Peer Review File · Nature Communications]

A multiple super-enhancer region establishes inter-TAD interactions and controls Hoxa function in cranial neural crestREVIEWER COMMENTS

Reviewer #1 (Remarks to the Author):

In this work, Kessler and colleagues made two contributions. On the one hand they describe thousands of Cranial Neural Crest Cells (CNCCs) Super Enhancers (SEs) and link them to putative cognate genes using promoter capture Hi-C (PCHi-C). On the other hand, they characterize the regulation of *Hoxa2* in CNCCs. They specifically demonstrate how *Hoxa2* is controlled by multiple SEs in an unusual way, as these enhancers appear to loop out from their own Topologically Associating Domain (TAD) to contact *Hoxa2*, which locates at the border of another neighboring TAD.

This study is a real tour de force. First the authors select and work with homogenous cell populations from micro-dissected embryonic organs (Md, PA2, Pinna) at different embryonic stages to perform a genome wide analysis of SEs, linking clusters of enhancers to their target gene by PCHi-C. This yields a useful resource to any ongoing and future studies on gene transcriptional control in these tissues. Second, the author characterizes in detail, again in sub-selected CNCCs, the role of two super-enhancer clusters (HIRE-1 and HIRE2) using state-of-the-art technologies. They show that both enhancer clusters can contact a target promoter, here *Hoxa2*, that locates outside of their TADs using Hi-C and PCHi-C. Using loss-of-function approaches in vivo they show that both enhancers' clusters are playing a role in controlling *Hoxa2* expression and ultimately in forming distinct parts of the external and middle ear. They finally reported that part of this regulation is dependent on *Hoxa2* in a typical feed forward fashion. Apart from the very clearcut results and elegant approach the authors display an in-depth knowledge of the literature which efficiently sets the study in proper context.

As this study is outstanding, this reviewer has only a few minor comments for the authors to answer.

1- It is unclear to this reviewer if HIRE-1 and 2 have potentially other target genes than *Hoxa2*. This question arises from the observation that in the Md tracks (Fig. 3 A) SE2/3/4 and SE5 seem to display both accessibility and H3K27ac signal even though the *HoxA* cluster is transcriptionally inactive. Also, are there other genes in HIRE-1/2 TAD or is this a gene-free TAD (which would be peculiar and noteworthy)? Using their HiC data, the authors could display virtual 4C profiles from the HIRE regions in the different tissues, similarly to what they did with the *Hoxa2* promoter in Fig. 3 C/D. They should also discuss this presence or absence of relevant intra-TAD target(s).

2- The authors could discuss the possibility of ectopic contacts between HIRE enhancer clusters and *Hoxa2* in tissues where the enhancers are active but not the gene. Such cases were already shown to be potentially detrimental (Deng et al., (Blobel lab), Kragestein and al., (Mundlos-Andrey lab)).

3- Although the authors already discuss it, it is important to bring even more forward in the conclusion that the position of *Hoxa2*, at a TAD border, could explain the peculiar inter-TAD enhancer-promoter contacts.

4- In Fig. 3, the C part could be aligned or have the same genomic coordinates as the A part for a better understanding of the signal over the locus. Moreover, in the D part the 3'TAD boundary position (peak of signal) could be highlighted.

5- Also, in Fig. 3, the HiC maps from sorted cell population are beautiful and convey very clearly the inter-TAD gain of contacts (stripes) between *Hoxa2* and HIREs. Yet, this reviewer would suggest the authors to produce subtraction maps to highlight in a very graphic manner the differences in structure between the different tissues.

6- The authors mention a dose-dependent effect of *Hoxa2* transcription loss on phenotypes. A figure that would put side to side, transcription loss and phenotype would be therefore useful to the reader.

7- Fig. 1 is not a great start in the paper as it is rather difficult to apprehend and could be re-thought to ease the reader understanding.

Reviewer #2 (Remarks to the Author):

Hoxa2 is a homeotic gene expressed in the rostral part of the mouse neural crest, particularly in the cells which migrate into the second pharyngeal arch. Genetic studies have shown that, in both mouse and human, mutations in Hoxa2 can cause defects in these second arch derivatives, especially with the bones of the middle ear and the external ear pinna (microtia). Several previous studies have examined proximal promoter and enhancer elements that can drive aspects of Hoxa2 expression in mouse, but none of these have been thoroughly tested for functionality by deletion of the endogenous sequences.

The current manuscript by Kebler et al addresses this problem by first identifying and analyzing super enhancer elements associated with cranial neural crest cell populations, following which they focus only on those sequences linked to the Hoxa2 promoter. Using a combination of their previously published histone methylation ChIP-seq data as well as Hi-C and promoter capture Hi-C they detail the presence of a group of super enhancers approximately 1MB upstream of the Hoxa2 promoter that sits on the other side of a TAD boundary. They divide these clustered super enhancers into two blocks, HIRE1 and HIRE2, based on their proximity to one another, as well as interactions with the Hoxa2 promoter in various craniofacial tissues over time. Both HIRE1 and HIRE2 interact with the Hoxa2 promoter in the pinna at E12.5 where Hoxa2 is expressed, but not in the E12.5 mandibular prominence where this gene is not expressed. They further show that elements within these super enhancers are shared at the sequence level between various vertebrate species from primates to fish. Next, they remove either the HIRE1 or HIRE2 block from the mouse genome using CRISPR/Cas9 and analyze the consequence on craniofacial development in comparison to the full Hoxa2 knockout mouse. Their data show that HIRE2, which contains a single super enhancer region, results in small effect on pinna size when it is combined in trans with a Hoxa2 null allele. In contrast, homozygous deletion of HIRE1, which contains three super enhancer regions, results in a significant loss of Hoxa2 expression and several phenotypes associated with Hoxa2 and Hoxa3 mutation.

All told the paper is very dense, detailed, and quite thorough in its analysis of the role that super enhancers can exert in directing craniofacial development. The data are generally strong and convincing with the first clear mechanistic insight into long range regulation of the Hoxa locus in vivo. In short, the manuscript provides important insight into a new aspect of craniofacial gene regulation, as well as how such super enhancers function in vivo to affect expression of several genes within the associated Hoxa locus. Despite the overall strengths of the study, there are still several areas that require additional information or analysis to ensure that the advances provided are exploited to their full potential.

Major issues

1.

In discussing the HIRE1 homozygous deletion, as well as mice containing one HIRE1 mutant allele and one Hoxa2 null allele, the authors state (page 8/9) that these mice phenocopy the skeletal defects seen in Hoxa2 null mice. They also mention that their new mouse strains die perinatally, but do not provide any observations why this might occur. Several previous studies have shown that Hoxa2 null mice have a high prevalence of cleft secondary palate that would lead to perinatal death. This result has often been considered a conundrum as Hoxa2 was not thought to be expressed more rostrally than the second pharyngeal arch. One hypothesis is that Hoxa2 is actually expressed later in the neural crest

derivatives of the palatal shelves and is responsible for their growth and fusion. Another theory held that defects in the hyoid arch caused defects in tongue movement such that it did not descend away from the roof of the mouth and so prevented palate fusion. The current manuscript should shed light on this issue but avoids doing so in its current form. They do not mention the secondary palate as a cause of lethality, and the skeletal images shown in Supplementary Figure 4 do not quite extend far enough towards the snout to assess the position of the palatine bones with respect to the midline. This issue is important as it addresses the range of craniofacial defects caused by loss of HIRE1 and if they do fully phenocopy the *Hoxa2* null phenotype as stated. Secondly, if the palate is closed it could indicate that HIRE1 does not regulate all *Hoxa2* expression in the rostral cranial neural crest.

2.

The authors indicate on page 5 that they have identified 2232 putative super enhancers, but they only show information on 177 of these. The authors should consider providing a supplementary table with information concerning the chromosomal location of all super enhancers they identified.

3.

On page 5 they mention performing “PCHi-C” but do not define what it means until later. A definition and a brief explanation of its purpose in the study might be helpful on page 5. In addition, though the Methods are generally very detailed, they should also provide information (e.g. source) concerning the biotinylated RNA fragments used for promoter capture so that coverage of craniofacial specific transcripts can be assessed.

4.

On page 11, the authors state in reference to Fig. 8 that “while in PA3, *Hoxa2* relative transcript levels decreased more than *Hoxa3*, as compared to WT”. Could the authors clarify that statement as it seems from Fig. 8C that *Hoxa3* has the bigger fold reduction. Note that Supp Fig 6 also has relevant information concerning this statement.

5.

On page 6, the authors noted that “PCHi-C revealed 5 putative SEs, SE1-5, selectively targeted *Hoxa2* in E10.5 PA2 CNCCs” referring to Fig. 1 and 3a. However, Fig 1 lists 6 *Hoxa2* interacting super enhancers. Five occur in a block in the middle of Fig 1, the sixth is nearer the bottom of the figure and is shared with *Hoxa3*. It is also notable that this interaction doesn’t seem to be present in Fig 3a. The reason for the exclusion of this sixth super enhancer from further analysis should be stated. On a separate note, the finding that several of the super enhancers connected with *Hoxa2* are also linked to *Hoxa3* might support the authors finding that *Hoxa3* expression levels are also down when HIRE1 is deleted.

Minor issues

1. Page 7. Referring to Fig 3a and Supp Fig. 3 “Notably, in Hox-negative Md, Mx and FNP CNCC populations HIRE1 and HIRE2 were also accessible”. No data are presented for Mx and FNP in these figures.

2. On page 11, and in Figure 8, the authors discuss genes whose expression is altered by loss of HIRE1. It would be helpful to know if any of these genes were closely linked to HIRE1 on chromosome 6 and if so, if any PCHi-C interactions were noticed with the associated promoters.

3. It would be useful to include any information on the sequence that exists at the site of HIRE1 and HIRE2 deletion following gene targeting if it is available.

4. The authors demonstrate a reduction of *Hoxa2* levels in the absence of HIRE1. If any analysis performed to assess changes in RNA expression when HIRE2 was deleted this should be mentioned.
5. Typo page 24 “until surround tissues” should be “surrounding”
6. Typo page 25 “The color reaction was initiated by placubg”.
7. Figure 3 legend. In reference to binding sites for *Hoxa2*, *Pbx* and *Meis* please note if these were derived by ChIP-seq or by sequence analysis.
8. Figure 7 and Supp Fig 6 legends, define “Md” on figure.
9. Figure 8 legend. “Heatmaps (b and d)... Replicates in PA2 (b) and PA3 (c).” Maybe that should be “PA2 (a)”.
10. Supp Fig 3 and legend. “From mandibular (Md) and second pharyngeal arch (PA2)”. No data for the Md is shown in Supp Fig 3.
11. Supp Table 1. “The table contains the genomic coordinates (mm10) of the 177 SEs”. The table has 183 rows and the difference in these two values (177 versus 183) should be explained.

Reviewer #3 (Remarks to the Author):

This is an exceptional study that integrates many types of epigenetic and 3D genome architecture datasets to uncover a cluster of long-range enhancers for the *Hoxa2* locus. *Hox* genes are the archetypes of embryonic positional patterning, and their complex regulation has through history provided new paradigms of how chromatin modifications and 3D genome architecture regulate gene expression. This study challenges the paradigm that topology associated domains (TADs) at as absolute barriers to enhancer-promoter interactions. Through a comprehensive identification of super-enhancers in the mouse pharyngeal arches, they show two such clusters of strong enhancers that lie outside the *Hoxa2* TAD yet have critical roles in its regulation. A strength of this study is that deletion of each cluster in mice results in distinct phenotypes of the facial skeleton and outer ear where *Hoxa2* is known to be essential. Whereas deletion of cluster HIRE1 replicates the homeotic phenotype of *Hoxa2* mutants, deletion of cluster HIRE2 causes more subtle defects in the outer ear. From a mechanistic angle, they show that while super-enhancers remain accessible in both *Hoxa2* negative and positive arches, lack of repressive chromatin in the second arch correlates with the ability of the *Hoxa2* locus to interact with the distant super-enhancers and be transcribed. They also show a novel autoregulatory mechanism whereby *Hoxa2* protein itself is required for the *Hoxa2* locus to interact with the late-acting outer ear HIRE2 cluster. The experiments are elegantly performed and precisely interpreted. This study will have a major impact in our understanding of how 3D genome architecture regulates key developmental patterning genes.

I only have minor suggestions of how the presentation of the work can be improved. None of these affect the major conclusions of this study.

1. In Supp. Table 1, it would be helpful to list which prominence (FNP, Mx, Md, PA2) the SE-gene interactions are predicted to more strongly occur.
2. In Fig. 1, why were the particular SEs/genes selected to be listed on the right column? Are

these the PA2 SEs or something else? This should be clarified in Legend.

3. More precise details of the exact HIRE1 and HIRE2 sequences deleted should be provided. For example, this could be diagrammed in Supp. fig. 3, including the genomic coordinates of the breakpoints of the deleted regions.

4. Fig. 7a,c could be presented better. The boxed genes hide the dots on the plots.

5. From Fig. 8d,e, it is hard to appreciate that Hoxa2 is more affected than Hoxa3 by HIRE1 deletion.

6. Fig. 9a-c feels a bit preliminary as this is a long list of related TF motifs and the exact TFs that bind these would require further validation. Can the major classes of TF motifs be condensed in the main figure, with more comprehensive lists of all predicted motifs moved to a table or supplemental figure? Also the cluster 1, 2, 3 categories are hard to appreciate in the main figure in that cluster 2 represents E10.5 enriched peaks, cluster 1 E12.5 peaks, and 3 both E10.5 and E12.5 peaks. It might be easier to follow if rather than listing cluster 1,2,3 in Fig. 9, the clusters are described by E10.5, E12.5, both – with rationale for this described elsewhere (e.g. existing supp. Fig.).

7. In Discussion, the paragraph on “domain-skipping interactions” was difficult to understand. This should be improved.

8. Several typos in Methods that should be copyedited. e.g. p. 25: “Then required amount”, “placubg the embryos”

POINT-BY-POINT RESPONSE TO REVIEWERS' COMMENTS NCOMMS-22-37040

We were delighted that all reviewers found this study of high significance (e.g. Reviewer #1: ...this study is **outstanding**..; Reviewer #2... the first clear **mechanistic insight into long range regulation** of the Hoxa locus *in vivo*; Reviewer #3....this is an **exceptional** study...). We thank all the reviewers for their insightful and constructive inputs which helped us to significantly improve our work. We have fully addressed all the reviewers' remarks with additional experiments, data analyses, revisions to figures, additional supplementary figures, and revisions to the text. We also added 3 figures to this letter just for the reviewers' perusal. Please find below a detailed account of the changes and experimental additions to the current revision of the manuscript. We have also accordingly revised the main text and improved discussion. We do hope that the reviewers will be fully satisfied by these revisions.

Reviewer #1 (Remarks to the Author):

In this work, Kessler and colleagues made two contributions. On the one hand they describe thousands of Cranial Neural Crest Cells (CNCCs) Super Enhancers (SEs) and link them to putative cognate genes using promoter capture Hi-C (PCHi-C). On the other hand, they characterize the regulation of Hoxa2 in CNCCs. The specifically demonstrate how Hoxa2 is controlled by multiple SEs in an unusual way, as these enhancers appear to loop out from their own Topologically Associating Domain (TAD) to contact Hoxa2, which locates at the border of another neighboring TAD.

*This study is a real tour de force. First the authors select and work with homogenous cell populations from micro-dissected embryonic organs (Md, PA2, Pinna) at different embryonic stages to perform a genome wide analysis of SEs, linking clusters of enhancers to their target gene by PCHi-C. This yields a useful resource to any ongoing and future studies on gene transcriptional control in these tissues. Second, the author characterize in detail, again in sub-selected CNCCs, the role of two super-enhancer clusters (HIRE-1 and HIRE2) using state-of-the-art technologies. They show that both enhancer clusters can contact a target promoter, here Hoxa2, that locates outside of their TADs using Hi-C and PCHi-C. Using loss-of-function approaches *in vivo* they show that both enhancers' clusters are playing a role in controlling Hoxa2 expression and ultimately in forming distinct parts of the external and middle ear. They finally reported that part of this regulation is dependent on Hoxa2 in a typical feed forward fashion. Apart from the very clearcut results and elegant approach the authors display an in-depth knowledge of the literature which efficiently sets the study in proper context.*

As this study is outstanding,...

We thank very much this reviewer for her/his very positive assessment.

...this reviewer has only a few minor comments for the authors to answer.

1- It is unclear to this reviewer if HIRE-1 and 2 have potentially other target gene than Hoxa2. This question arises from the observation that in the Md tracks (Fig. 3A) SE2/3/4 and SE5 seem to display both accessibility and H3K27ac signal even though the HoxA cluster is transcriptionally inactive. Also, are there other genes in HIRE-1/2 TAD or is this a gene-free TAD (which would be peculiar and noteworthy)? Using their HiC data, the authors could display virtual 4C profiles from the HIRE regions in the different tissues, similarly to what they did with the Hoxa2 promoter in Fig. 3C/D. They should also discuss this presence or absence of relevant intra-TAD target(s).

Response: The TAD containing HIRE1 and HIRE2 contains only very few genes (e.g. *Npvf*; please see annotation in figure 3c). However, HIRE1 and HIRE2 do not interact with any other expressed gene within or outside their own TAD besides *Hoxa2* in PA2 (e.g. **Figure R1** below, just for the reviewer's perusal, shows that *Npvf* is not expressed in any of the analyzed CNCC subpopulations). This indicates that HIRE1 and HIRE2 are *Hoxa2*-specific. As suggested by the reviewer we display virtual 4C plots from the HIRE1 or HIRE2 viewpoints, confirming the *Hoxa2*-specific interactions (**new** Suppl. fig. 4).

Figure R1

We have accordingly revised the text as follows (lines **197-205** of edited version):

'The TAD containing HIRE1 and HIRE2 is a gene-poor chromosomal region (Fig. 3c). Virtual 4C plots from the HIRE1 or HIRE2 viewpoints confirmed the Hoxa2-specific interactions. Furthermore, they identified weak intra- and inter-TAD interactions with Npvf and Hoxa1/Hoxa3, respectively (Suppl. fig. 4). None of these genes are expressed in the selected CNCC subpopulations, suggesting that these weak contacts are not functional nor specific, likely due to physical proximity of Hoxa1/Hoxa3 to Hoxa2 and Npvf to the HIRE1/2 SEs, respectively. Moreover, most HiC-based methods cannot resolve very proximal interactions (typically < 30kb) from the generally high background crosslinking frequency between genomic sequences over these distances.'

2- The authors could discuss the possibility of ectopic contacts between HIRE enhancer clusters and Hoxa2 in tissues where the enhancers are active but not the gene. Such cases where already shown to be potentially detrimental (Deng et al., (Blobel lab), Kragestein and al, (Mundlos-Andrey lab)).

Response: We have revised the Discussion as follows (lines **544-567** of edited version, revised text added in *italics*):

Intriguingly, HIRE1 and HIRE2 are active, i.e. enriched with H3K27ac, both in *Hoxa2*-expressing PA2 and *Hox*-free Md CNCCs, yet the contacts between HIRE1 and HIRE2 and *Hoxa2* only occur in PA2 CNCCs (Fig. 3a, 3b). In *Hox*-free CNCCs, the *Hoxa2* promoter may not be available for HIRE1/HIRE2 interaction since, together with the whole *Hoxa* cluster, it is embedded in a large repressive Polycomb domain¹⁶ which may segregate in a repressive nuclear compartment distinct from the active HIRE1/HIRE2. *Indeed, Polycomb binding at Hoxa promoters in developing limbs can prevent their interaction with active enhancers*⁹⁹. In PA2 CNCCs, local patterning signals may activate proximal enhancers so that the *Hoxa2* locus is 'singled out' from the Polycomb repressed *Hoxa* cluster and transcriptionally induced, followed by removal of H3K27me3 and switch to H3K27ac deposition¹⁶; this might in turn allow rapid interaction with the inter-TAD SEs boosting *Hoxa2* expression to full transcriptional output.

Similarly, *Hoxa3* might become collinearly connected to HIRE1/HIRE2 in PA3 CNCCs, where *Hoxa3* is transcriptionally induced and *Hoxa2* is expressed as well.

Furthermore, tissue-specific 3D chromatin conformation can also contribute to enhancer activity and specificity⁶. For instance, the Pen enhancer shows activity in both developing forelimbs and hindlimbs, but it only controls Pitx1 transcription in hindlimbs. This restricted enhancer activity is associated with a 3D chromatin configuration allowing Pen and Pitx1 to interact only in hindlimbs, whereas enhancer and promoter are maintained physically separated in forelimbs⁶. Structural chromatin variants can however convert the inactive into an active 3D conformation, thereby inducing Pitx1 misexpression in forelimbs⁶. Moreover, forced chromatin looping of strong enhancers to developmentally silenced promoters can be sufficient to stimulate transcription^{100,101}. Thus, HIRE1/HIRE2-driven transcriptional regulation of Hoxa2 may be allowed by a PA2-specific 3D chromatin configuration.

3- Although the authors already discuss it, it is important to bring even more forward in the conclusion that the position of Hoxa2, at a TAD border, could explain the peculiar inter-TAD enhancer-promoter contacts.

Response: As pointed out by the reviewer, this speculative point was already discussed in the previous version. We have now revised it in the Discussion to bring it more clearly forward as follows (lines **567-575** of edited version):

"Moreover, HIRE1 and HIRE2 might be brought in proximity to the *Hoxa2* locus by a mechanism similar to the domain-skipping interactions described between *Drosophila Scr* and its distal enhancer T1, whereby formation of an intervening TAD by boundary pairing is essential for distal, inter-TAD, E-P interaction¹⁰². Active *Hoxa2* and its SE region HIRE1/HIRE2 are both close to interacting TAD boundaries (Fig. 3b), suggesting that pairing between boundary elements might bring distant HIRE1/HIRE2 and its target promoter in proximity by domain-skipping chromatin folding in PA2 CNCCs."

4- In Fig. 3, the C part could be aligned or have the same genomic coordinates as the A part for a better understanding of the signal over the locus. Moreover, in the D part the 3'TAD boundary position (peak of signal) could be highlighted.

Response: In revised fig. 3, we have now used the same genomic coordinates for plots a and c, and we have aligned these 2 plots. We have highlighted the TAD boundary in fig. 3c. Panel 3d is a zoom-in on virtual 4C profiles from panel 3c at HIRE1 and HIRE2 (chr6:50779503–51183201); the TAD boundary is not present.

5- Also, in Fig. 3, the HiC maps from sorted cell population are beautiful and convey very clearly the inter-TAD gain of contacts (stripes) between Hoxa2 and HIREs. Yet, this reviewer would suggest the authors to produce subtraction maps to highlight in a very graphic manner the differences in structure between the different tissues.

Response: As suggested by the reviewer, we have now produced HiC subtraction heat maps, which we present in the **new** suppl. Fig. 4, namely between:

1. The mandibular process (Md) and second pharyngeal arch (PA2)-derived cranial neural crest cells (CNCCs) at E10.5;
2. The pinna-derived CNCCs at E12.5 and PA2-derived CNCCs at E10.5;
3. The pinna-derived CNCCs at E12.5 and E14.5.

6- The authors mention a dose-dependent effect of Hoxa2 transcription loss on phenotypes. A figure that would put side to side, transcription loss and phenotype would be therefore useful to the reader.

Response: As suggested by the reviewer, we now provide a **new** supplementary figure (suppl. fig. 10) where we put side to side in a diagrammatic representation the different extents of *Hoxa2* transcript reduction/loss induced by distinct mutations at E10.5, and the corresponding phenotypes of PA2 derived cartilaginous and skeletal structures.

7- Fig. 1 is not a great start in the paper as it is rather difficult to apprehend and could be re-thought to ease the reader understanding.

Response: We agree and thank the reviewer for this comment. We have now revised figure 1 to improve its understanding. In Fig. 1a, we added a schematic diagram representing the mouse facial prominences in different colors at embryonic day 10.5 (E10.5), in order to highlight the cranial neural crest cells (CNCCs) sub-populations where the super-enhancers (SE) have been identified. We have further added in fig. 1a representative examples of ChIP-seq and RNA-seq tracks and PChI-C connectivity pattern at the level of a promoter-SE pairs where both elements are active. To improve understanding of the data, we have also simplified the heat map (Fig. 1b). As we have now added an additional criterion for the identification of SEs which includes the expression of the target genes in the subpopulation(s) where the SE is active and connected to its promoter (please see main text lines **124-137** of edited version), we have now 148 unique promoter-SE pairs (rows) instead of 183 before. Moreover, the row annotations now highlight more transcription factor coding genes that have been associated with craniofacial development and/or malformations.

Reviewer #2 (Remarks to the Author):

Hoxa2 is a homeotic gene expressed in the rostral part of the mouse neural crest, particularly in the cells which migrate into the second pharyngeal arch. Genetic studies have shown that, in both mouse and human, mutations in Hoxa2 can cause defects in these second arch derivatives, especially with the bones of the middle ear and the external ear pinna (microtia). Several previous studies have examined proximal promoter and enhancer elements that can drive aspects of Hoxa2 expression in mouse, but none of these have been thoroughly tested for functionality by deletion of the endogenous sequences.

The current manuscript by Kebler et al addresses this problem by first identifying and analyzing super enhancer elements associated with cranial neural crest cell populations, following which they focus only on those sequences linked to the Hoxa2 promoter. Using a combination of their previously published histone methylation ChIP-seq data as well as Hi-C and promoter capture Hi-C they detail the presence of a group of super enhancers approximately 1MB upstream of the Hoxa2 promoter that sits on the other side of a TAD boundary. They divide these clustered super enhancers into two blocks, HIRE1 and HIRE2, based on their proximity to one another, as well as interactions with the Hoxa2 promoter in various craniofacial tissues over time. Both HIRE1 and HIRE2 interact with the Hoxa2 promoter in the pinna at E12.5 where Hoxa2 is expressed, but not in the E12.5 mandibular prominence where this gene is not expressed. They further show that elements within these super enhancers are shared at the sequence level between various vertebrate species from primates to fish. Next, they remove either the HIRE1 or HIRE2 block from the mouse genome using CRISPR/Cas9 and analyze the consequence on craniofacial development in comparison to the full Hoxa2 knockout mouse. Their data show that HIRE2, which contains a single super enhancer region, results in small effect on pinna size when it is combined in trans with a Hoxa2 null allele. In contrast, homozygous deletion of HIRE1, which contains three super enhancer regions, results in a significant loss of Hoxa2 expression and several phenotypes associated with Hoxa2 and Hoxa3 mutation.

All told the paper is very dense, detailed, and quite thorough in its analysis of the role that super enhancers can exert in directing craniofacial development. The data are generally strong and convincing with the first clear mechanistic insight into long range regulation of the *Hoxa* locus *in vivo*. In short, the manuscript provides important insight into a new aspect of craniofacial gene regulation, as well as how such super enhancers function *in vivo* to affect expression of several genes within the associated *Hoxa* locus.

We thank this reviewer for her/his very positive assessment.

Despite the overall strengths of the study, there are still several areas that require additional information or analysis to ensure that the advances provided are exploited to their full potential.

Major issues

1. In discussing the *HIRE1* homozygous deletion, as well as mice containing one *HIRE1* mutant allele and one *Hoxa2* null allele, the authors state (page 8/9) that these mice phenocopy the skeletal defects seen in *Hoxa2* null mice. They also mention that their new mouse strains die perinatally, but do not provide any observations why this might occur. Several previous studies have shown that *Hoxa2* null mice have a high prevalence of cleft secondary palate that would lead to perinatal death. This result has often been considered a conundrum as *Hoxa2* was not thought to be expressed more rostrally than the second pharyngeal arch. One hypothesis is that *Hoxa2* is actually expressed later in the neural crest derivatives of the palatal shelves and is responsible for their growth and fusion. Another theory held that defects in the hyoid arch caused defects in tongue movement such that it did not descend away from the roof of the mouth and so prevented palate fusion. The current manuscript should shed light on this issue but avoids doing so in its current form. They do not mention the secondary palate as a cause of lethality, and the skeletal images shown in Supplementary Figure 4 do not quite extend far enough towards the snout to assess the position of the palatine bones with respect to the midline. This issue is important as it addresses the range of craniofacial defects caused by loss of *HIRE1* and if they do fully phenocopy the *Hoxa2* null phenotype as stated. Secondly, if the palate is closed it could indicate that *HIRE1* does not regulate all *Hoxa2* expression in the rostral cranial neural crest.

Response: We thank the reviewer for this comment which gives us the opportunity to provide additional data and description. Of course, we are fully aware of the cleft palate phenotype, which we firstly described in our original *Hoxa2* knockout paper about 30 years ago (Rijli et al., Cell 1993). We have now added information regarding the secondary palate cleft in *HIRE1*^{del/del} and *HIRE1*^{del/wt};*Hoxa2*^{EGFP/wt} mutants fetuses and observed that these mutants display secondary palate cleft in the same range as the full *Hoxa2* mutants.

We have accordingly generated a **new** suppl. figure (Suppl. Fig. 6) and revised the main text as follows (lines **254-257** of edited version):

"Similar to *Hoxa2*^{EGFP/EGFP} 49 mutants (n= 6/8), both *HIRE1*^{del/del} and *HIRE1*^{del/wt};*Hoxa2*^{EGFP/wt} mutants also displayed a cleft secondary palate (n = 8/10 and 14/24, respectively) (Suppl. fig. 6)."

We also provide below, just for this reviewer's perusal, a figure (**Figure R2**) showing that *Hoxa2* is not expressed in the developing palatal shelves. To evaluate the presence of *Hoxa2*-expressing cells to the palatal shelves, we used our previously published *Hoxa2*^{EGFP} knockin allele (Pasqualetti et al., 2002) which provides an even better readout than *in situ* hybridisation, given the EGFP stability. We performed DAPI staining and EGFP immunostaining on coronal sections of E12.5 and E14.5 *Hoxa2*^{EGFP} embryos. No EGFP positive cells were observed in the developing palatal shelves, indicating absence of *Hoxa2* expression.

Palatal shelves

(A-H) Dapi staining (blue) and anti-EGFP immunostaining (green) on coronal sections of E12.5 (A-D) and E14.5 (E-H) $Hoxa2^{EGFP/WT}$ embryos, showing the palatal shelves (PS) before (E12.5) and after (E14.5) their fusion. C and D are enlarged views of the PS shown in A and B respectively, while G and H are enlarged views of the PS shown in G and H respectively. Note that A and E are more anterior than B and F respectively.

Figure R2

2. The authors indicate on page 5 that they have identified 2232 putative super enhancers, but they only show information on 177 of these. The authors should consider providing a supplementary table with information concerning the chromosomal location of all super enhancers they identified.

Response: We now provide a **new** supplementary table (suppl. table 1) which contains the genomic locations (mm10) of all super-enhancers identified in CNCC subpopulations at E10.5 based on their H3K27ac signal.

3. On page 5 they mention performing "PCHi-C" but do not define what it means until later. A

definition and a brief explanation of its purpose in the study might be helpful on page 5. In addition, though the Methods are generally very detailed, they should also provide information (e.g. source) concerning the biotinylated RNA fragments used for promoter capture so that coverage of craniofacial specific transcripts can be assessed.

Response: We have accordingly revised the main text and Methods as follows (added text is in italics):

Main text (lines **129-134** of edited version): " To this aim, we performed PCHi-C³⁷ in duplicate for each of the four CNCC subpopulations (Suppl. fig. 2a). *Briefly, we used biotinylated RNA bait probes targeting promoter regions (methods) to selectively enrich for all distal genome-wide sequences interacting with promoters from a pool of ‘all-to-all’ genomic interactions generated by Hi-C, followed by high throughput paired-end sequencing and statistical analysis.*"

Methods (lines **689-692** of edited version):" Promoter capture Hi-C was performed as described previously¹⁰⁷ with adaptations to our system. *39021 biotinylated RNA probes were designed to enrich for 22,225 annotated gene promoters in the mouse genome using PCHi-C. These probes were designed and used as described in^{37,107}, and purchased from Agilent Technologies.*"

4. On page 11, the authors state in reference to Fig. 8 that “while in PA3, Hoxa2 relative transcript levels decreased more than Hoxa3, as compared to WT”. Could the authors clarify that statement as it seems from Fig. 8C that Hoxa3 has the bigger fold reduction. Note that Supp Fig 6 also has relevant information concerning this statement.

Response: We thank the reviewer for bringing this to our attention. It was a mistake in the text. In Fig. 8d, 8e, we can indeed see that in PA3, *Hoxa2* is less affected than *Hoxa3* by HIRE1 deletion. We amended the text (lines **339-343** of edited version).

5. On page 6, the authors noted that “PCHi-C revealed 5 putative SEs, SE1-5, selectively targeted Hoxa2 in E10.5 PA2 CNCCs” referring to Fig. 1 and 3a. However, Fig. 1 lists 6 Hoxa2 interacting super enhancers. Five occur in a block in the middle of Fig. 1, the sixth is nearer the bottom of the figure and is shared with Hoxa3. It is also notable that this interaction doesn’t seem to be present in Fig 3a. The reason for the exclusion of this sixth super enhancer from further analysis should be stated. On a separate note, the finding that several of the super enhancers connected with Hoxa2 are also linked to Hoxa3 might support the authors finding that Hoxa3 expression levels are also down when HIRE1 is deleted.

Response: We thank the reviewer for this comment. Indeed, our PCHi-C experiments revealed the presence of 5 SEs, SE1-5, selectively targeting *Hoxa2* in E10.5 PA2 CNCCs. We also observed the presence of a sixth super-enhancer (SE6; genomic coordinates: chr6:51579396–51613150). This is highlighted in yellow in the figure provided below for this reviewer's perusal, **Figure R3**. SE6 shows a significant interaction (CHiCAGO score>5) with the *Hoxa2* promoter in FNP and Mx only, where this gene is not expressed and inactive (embedded in a Polycomb repressive domain, Minoux *et al.*, 2017). When investigating this interaction further, we found that only 1 out of 11 restriction fragments (chr6:51589493-51591036, highlighted in blue) overlapping with SE6 shows a significant interaction (CHiCAGO score of 5.86 for FNP and of 5.25 for Mx) with the *Hoxa2* promoter. This restriction fragment is located between two CTCF sites, which may explain the observed contact with the *Hoxa2* promoter (see CTCF ChIP-seq tracks in **Figure R3** and suppl. fig. 2). Thus, for downstream analysis, we decided to focus only on SE1-5, that selectively target *Hoxa2* in the CNCCs in which it is expressed, i.e. in PA2. As the 6th SE region does not seem to act as a super-enhancer for *Hoxa2* expression in PA2, it was not further considered in downstream analysis.

Figure R3

Furthermore, the comment of this reviewer helped us to improve the methodology and introduce an additional criterion for selection of putative SEs. Indeed, the SE definition relies purely on the enrichment of H3K27ac and does not consider the functional impact of the SE on its target gene's transcription. Since interaction of a gene with a (super-)enhancer is expected to increase its target gene's transcription, we decided to apply an additional criterion when identifying the SEs linked to the differentially expressed transcription factor coding genes displayed in the heatmap of fig. 1. Namely, the SE should be active (i.e. above the relative cutoff) in the same subpopulation as the significant link to the target gene's promoter is observed and the target gene should be expressed there as well, with an RPKM>2.

We have now added this information in the results section (lines **124-137** of edited version) and in the legend of fig. 1. With this additional criterion we identified 147 SEs (instead of 177 in the first version). Among the putative SEs that are not anymore in the list is SE6 that weakly targets *Hoxa2* in Mx and FNP. The putative SEs that targeted *Hoxa3* in PA2 are also not any longer in the list, the *Hoxa3* transcript being below the threshold of detection.

Minor issues

1. Page 7. Referring to Fig 3a and Supp Fig. 3 “Notably, in Hox-negative Md, Mx and FNP CNCC populations HIRE1 and HIRE2 were also accessible”. No data are presented for Mx and FNP in these figures.

Response: We have now added the ATAC-seq data for FNP, Mx and Md in supplementary figure 3a.

2. On page 11, and in Figure 8, the authors discuss genes whose expression is altered by loss of HIRE1. It would be helpful to know if any of these genes were closely linked to HIRE1 on chromosome 6 and if so, if any PChI-C interactions were noticed with the associated promoters.

Response: We thank the reviewer for this comment, which we have thoroughly addressed above in our response to point 1 of Reviewer 1. Please refer to that response.

3. It would be useful to include any information on the sequence that exists at the site of HIRE1 and HIRE2 deletion following gene targeting if it is available.

Response: We have now added this information in **new** suppl. fig. 3b.

4. The authors demonstrate a reduction of Hoxa2 levels in the absence of HIRE1. If any analysis performed to assess changes in RNA expression when HIRE2 was deleted this should be mentioned.

Response: We thank the reviewer for this comment. We have now performed a qPCR analysis in PA2 of E10.5 $HIRE2^{del/del}$, $HIRE2^{del/wt};Hoxa2^{EGFP/wt}$, $HIRE1^{del/del}$ and $HIRE1^{del/wt};Hoxa2^{EGFP/wt}$ mutants embryos (**new** Suppl. fig. 9). We collected PA2 of E10.5 WT and $Hoxa2^{EGFP/wt}$ embryos as controls (new Suppl. fig. 9). In $Hoxa2^{EGFP/wt}$ embryos, we observed, as expected, about 50% of $Hoxa2$ transcripts as compared to WT (new Suppl. fig. 9). In $HIRE2^{del/del}$ and $HIRE2^{del/wt};Hoxa2^{EGFP/wt}$ mutant embryos, we found respectively 75% and 46% of $Hoxa2$ transcripts as compared to WT (new Suppl. fig. 9), which is in keeping with the absence of visible malformations at PA2-derived structures in E18.5 $HIRE2^{del/del}$ mutant fetuses, and the presence of mild malformations and in E18.5 $HIRE2^{del/wt};Hoxa2^{EGFP/wt}$ fetuses and microtia in adult mutants (**new** Suppl. fig. 10).

We have accordingly revised the text (lines **306-321** of edited version).

5. Typo page 24 “until surround tissues” should be “surrounding”

OK. Thank you.

6. Typo page 25 “The color reaction was initiated by placubg”.

OK. Thank you.

7. Figure 3 legend. In reference to binding sites for Hoxa2, Pbx and Meis please note if these were derived by ChIP-seq or by sequence analysis.

Response: The binding sites for Hoxa2, Pbx and Meis have been identified by ChIP-seq (Donaldson et al. 2012, Amin et al. 2015). We have added this information in the legend of figs. 3 and 9 and suppl. fig. 3.

8. Figure 7 and Supp Fig 6 legends, define “Md” on figure.

Response: We have now defined “Md” in the legend of fig. 7 and suppl. fig. 8 (formerly suppl. fig. 6).

9. Figure 8 legend. “Heatmaps (b and d).... Replicates in PA2 (b) and PA3 (c).” Maybe that should be “PA2 (a)”.

OK. Thank you.

10. Supp Fig 3 and legend. “From mandibular (Md) and second pharyngeal arch (PA2)”. No data for the Md is shown in Supp Fig 3.

Response: We have now added the ATAC-seq data for FNP, Mx and Md in suppl. fig. 3a.

11. Supp Table 1. “The table contains the genomic coordinates (mm10) of the 177 SEs”. The table has 183 rows and the difference in these two values (177 versus 183) should be explained.

Response: We have now considered an additional criterion for the selection of SEs based on the expression of the target genes in the subpopulation(s) where the SE is active and connected to its promoter (please see above, response to point 5).

With this additional criterion, we now identified 147 SEs instead of 177. The supplementary table 2 (formerly suppl. table 1) contains 148 unique pairs (rows) because one SE is shared by 2 different promoters (*Twist2*, *Hes6*) and thus it appears twice.

We have added this information in the Results section (lines **144-148** of edited version) and in the legend of fig. 1.

In the legend of suppl. table 2 (formerly suppl. table 1) we have added the following explanation for the differences between the number of SEs identified (147) and the number of rows present on the table (148) : “Note that the table contains 148 unique SE-promoter pairs (rows) because one SE is shared by 2 different promoters (*Twist2* and *Hes6*), and thus it appears twice”.

Reviewer #3 (Remarks to the Author):

This is an exceptional study..

We thank very much this reviewer for her/his very positive assessment.

..that integrates many types of epigenetic and 3D genome architecture datasets to uncover a cluster of long-range enhancers for the Hoxa2 locus. Hox genes are the archetypes of embryonic positional patterning, and their complex regulation has through history provided new paradigms of how chromatin modifications and 3D genome architecture regulate gene expression. This study challenges the paradigm that topology associated domains (TADs) at as absolute barriers to enhancer-promoter interactions. Through a comprehensive identification of super-enhancers in the mouse pharyngeal arches, they show two such clusters of strong enhancers that lie outside the Hoxa2 TAD yet have critical roles in its regulation. A strength of this study is that deletion of each cluster in mice results in distinct phenotypes of the facial skeleton and outer ear where Hoxa2 is known to be essential. Whereas deletion of cluster HIRE1 replicates the homeotic phenotype of Hoxa2 mutants, deletion of cluster HIRE2 causes more subtle defects in the outer ear. From a mechanistic angle, they show that while super-enhancers remain accessible in both Hoxa2 negative and positive arches, lack of repressive chromatin in the second arch correlates with the ability of the Hoxa2 locus to interact with the distant super-enhancers and be transcribed. They also show a novel autoregulatory mechanism whereby Hoxa2 protein itself is required for the Hoxa2 locus to interact with the late-acting outer ear HIRE2 cluster. The experiments are elegantly performed and precisely interpreted. This study will have a major impact in our

understanding of how 3D genome architecture regulates key developmental patterning genes.

I only have minor suggestions of how the presentation of the work can be improved. None of these affects the major conclusions of this study.

1. In Suppl. Table 1, it would be helpful to list which prominence (FNP, Mx, Md, PA2) the SE-gene interactions are predicted to more strongly occur.

Response: In suppl. table 2 (formerly suppl. table 1), we have now added the interaction strengths of each SE-promoter pairs in the frontonasal process (FNP), mandibular (Md), maxillary (Mx) and second pharyngeal arch (PA2) cranial neural crest cell subpopulations.

2. In Fig. 1, why were the particular SEs/genes selected to be listed on the right column? Are these the PA2 SEs or something else? This should be clarified in Legend.

Response: The gene names that were present on the right part of the heat map in former Fig. 1 were selected as a few known examples of genes involved in craniofacial development and/or whose mutations result in craniofacial anomalies.

We have now extended this list and included even more genes for transcription factors known to be important in these processes. We clarified it in the legend of figure 1b: “The row annotation highlights transcription factor coding genes that have been previously associated with craniofacial development and/or malformations”.

3. More precise details of the exact HIRE1 and HIRE2 sequences deleted should be provided. For example, this could be diagrammed in Suppl. fig. 3, including the genomic coordinates of the breakpoints of the deleted regions.

Response: We have now added this information in **new** suppl. figure 3b.

4. Fig. 7a,c could be presented better. The boxed genes hide the dots on the plots.

Response: We thank the reviewer for this comment. We guess the reviewer was referring to fig. 8. We have now improved the presentation of the up- and down-regulated genes upon HIRE1 deletion in PA2 and PA3, in fig. 8a, 8c.

5. From Fig. 8d,e, it is hard to appreciate that Hoxa2 is more affected than Hoxa3 by HIRE1 deletion.

Response: We thank the reviewer for bringing this to our attention. It was a mistake in the text. In Fig. 8d, 8e, we can indeed see that in PA3, *Hoxa2* is less affected than *Hoxa3* by HIRE1 deletion. We amended the text (lines **339-343** of edited version).

6. Fig. 9a-c feels a bit preliminary as this is a long list of related TF motifs and the exact TFs that bind these would require further validation. Can the major classes of TF motifs be condensed in the main figure, with more comprehensive lists of all predicted motifs moved to a table or supplemental figure? Also the cluster 1, 2, 3 categories are hard to appreciate in the main figure in that cluster 2 represents E10.5 enriched peaks, cluster 1 E12.5 peaks, and 3 both E10.5 and E12.5 peaks. It might be easier to follow if rather than listing cluster 1,2,3 in Fig. 9, the clusters are described by E10.5, E12.5, both – with rationale for this described elsewhere (e.g. existing suppl. Fig.).

Response: We agree with the reviewer. As suggested, we simplified the description and condensed the major classes of TF motifs in figure 9a, while the comprehensive list of all predicted motifs were moved to several **new** suppl. figures and **new** suppl. tables i.e. suppl. fig. 11 - formerly suppl. fig. 7, suppl. fig. 12, suppl. table 4, suppl. table 5 and suppl. table 6.

Briefly, through additional computational analysis steps (revised and detailed in Methods; lines **1011-1024** of edited version), we restricted the number of significantly enriched TF motifs from formerly 382 to 107 TF motifs with a high predicted hit rate in the 31 E10.5 and E12.5 ATAC-seq peaks within the HIRE1/HIRE2 region (**new** suppl. fig. 11g, **new** suppl. table 5). These motifs were subdivided into 10 clusters by motif similarity analysis (**new** suppl. Fig. 12, **new** suppl. fig. 11g, **new** suppl. table 6). Except for motif similarity cluster 7, the other nine clusters contained very similar motifs and were enriched for certain families of binding factors. One representative motif from each of these nine clusters was selected to be depicted in Figure 9a, including the name of the transcription factor family predicted to bind the corresponding cluster sequences. The full list of TF motifs in each of the ten clusters can be found in suppl. fig. 11g.

We have then accordingly revised the main text as follows (lines **372-405** of edited version):

"To investigate which transcription factors may be involved in binding HIRE1 and HIRE2 and potentially regulate long-range interactions with *Hoxa2*, we performed a motif enrichment analysis using the ATAC-seq data from PA2 and pinna CNCCs at E10.5 and E12.5, respectively. We first called the peaks from both stages, merged them, and extracted putative enhancer peaks to obtain a total of 106,587 peaks. 31 ATAC-seq peaks overlapped the region spanning from HIRE2 to HIRE1 (mouse GRCm38/mm10 chr6:50,789,172-51,087,888) (Suppl. fig. 11a). We then clustered these 31 peaks according to their relative accessibility at E10.5 and E12.5, (cluster 1-3) resulting in three separate clusters (cluster 1-3) (Suppl. fig. 11b). Cluster 1 peaks were more accessible at E12.5, cluster 2 peaks were more accessible at E10.5, whereas cluster 3 peaks only showed minor accessibility differences between E10.5 and E12.5.

Next, assuming that the observed accessibility changes are driven by differential transcription factor binding and that other genomic regions bound by the same transcription factors would show similar accessibility profiles, we ranked all ATAC peaks in E10.5 PA2 and E12.5 pinna CNCCs according to their similarity (Pearson's correlation coefficient) to the average accessibility profile of each of the three clusters (Suppl. fig. 11c). We selected the 1000 peaks with the highest correlation to each cluster profile, resulting in three non-overlapping peak sets (Suppl. fig. 11d). We then used these three sets of peaks and, as a control, a fourth set containing all residual ATAC peaks, and ran a motif enrichment analysis⁶² resulting in a total of 382 significantly enriched motifs (Suppl. fig. 11e; Suppl. table 4). (Suppl. fig. 7d). From these, we hierarchically clustered similar motifs and selected nine representative motifs with predicted transcription factor binding sites in the 31 ATAC-seq peaks overlapping with the HIRE2/HIRE1 region (Fig. 9a; Suppl. table 5; Suppl. table 6; Suppl. fig. 11g; Suppl. fig. 12)(methods).

For example, among the representative motifs there were binding sites for Tal- and NFAT-related factors, potentially bound by Twist1, Twist2, ZBTB18 and Nfatc1, Nfat5 and Nfatc3, respectively (Fig. 9a; Suppl. fig. 11g). Each of these transcription factors is upregulated from E10.5 to E12.5 in CNCCs (Suppl. table 3). Furthermore, we identified motifs for Hox-related factors, such as *Hoxa2*, and TALE-type homeodomain factors, such as Pbx and Meis. Both Meis and Pbx transcription factors are known Hox cofactors and form heterodimers with Hox proteins to bind to DNA^{63,64}. Analysis of published ChIP-seq datasets for *Hoxa2*, Pbx, and Meis in PA2 at E11.5^{57,58}, revealed that these factors indeed showed enriched binding at HIRE1 and HIRE2 (Fig. 3a; Suppl. fig. 3)."

7. In Discussion, the paragraph on "domain-skipping interactions" was difficult to understand. This should be improved.

Response: We have now revised it in the Discussion to bring it more clearly forward as follows (lines **567-575** of edited version):

"Furthermore, HIRE1 and HIRE2 might be brought in proximity to the *Hoxa2* locus by a mechanism similar to the domain-skipping interactions described between *Drosophila Scr* and its distal enhancer T1, whereby formation of an intervening TAD by boundary pairing is essential for distal, inter-TAD, E-P interaction¹⁰². Active *Hoxa2* and its SE region HIRE1/HIRE2 are both close to interacting TAD boundaries (Fig. 3b), suggesting that pairing between boundary elements might bring distant HIRE1/HIRE2 and its target promoter in proximity by domain-skipping chromatin folding in PA2 CNCCs."

8. Several typos in Methods that should be copyedited. e.g. p. 25: "Then required amount", "placubg the embryos"

OK. Thank you.

REVIEWERS' COMMENTS

Reviewer #1 (Remarks to the Author):

The authors have satisfactorily answered to all the points raised in the first round. Therefore, this reviewer recommends the manuscript for publication.

Reviewer #2 (Remarks to the Author):

The authors have nicely addressed all the comments in the previous critiques. This is a very thorough and interesting paper that deserves to be widely seen. The authors should be very proud of this exceptional work.

Reviewer #3 (Remarks to the Author):

This revision addresses all my concerns, which were minor. It remains an excellent study that should be of wide interest to developmental biologists.